# Open chromatin-guided interpretable machine learning reveals cancer-specific chromatin features in cell-free DNA
Sakuntha D. Gunarathna [1], Aerica Nagornyuk[1], Nazim A. Belabbaci [1], Regina Nguyen[1], Bappa Ghosh [1], Sabha Ganai[2], Tabatha Lemke[3], Cassy Garry[3], Mika Saotome[1], Muhan Yu[4], Mamoru Takada[4], Xusheng Wang [5] & Motoki Takaku [1] ✉

Cell-free DNAs (cfDNAs) are DNA fragments found in blood, originating mainly from immune cells in healthy individuals and from both immune and cancer cells in cancer patients. While cancer-derived cfDNAs carry mutations, they also retain epigenetic features such as DNA methylation and nucleosome positioning. In this study, we examine nucleosome enrichment patterns in cfDNAs from breast and pancreatic cancer patients and find significant enrichment at open chromatin regions. Differential enrichment is observed not only at cancer cell type specific ATAC-seq peaks but also at CD4$^+$ T cell specific peaks, suggesting both tumor- and immune-derived contributions to the cfDNA signal. To leverage these patterns, we apply an interpretable machine learning model (XGBoost) trained on cell type specific open chromatin regions. This approach improves cancer detection accuracy and highlights key genomic loci associated with the disease state. Our pipeline provides a robust and interpretable framework for cfDNA-based cancer detection.

Early cancer detection is key to reducing the mortality rate[1]. Liquid biopsy has emerged as a noninvasive diagnostic approach for detecting cancer-specific markers, predominantly proteins[2]. Recent findings have expanded its utility to the analysis of nucleic acids in samples, providing a new avenue for early cancer detection in a noninvasive manner. Liquid biopsy samples such as plasma, serum, and urine have been found to contain both DNAs and RNAs[3,4]. When samples are collected from donors with specific health conditions, such as cancer, chronic diseases, or pregnancy, signs related to these conditions can be detected[5,6]. For instance, circulating cell-free DNAs (cfDNA) purified from cancer patients include a fraction derived from tumors (circulating tumor DNAs or ctDNAs)[7]. The analysis of cfDNA sequences can reveal genetic alterations such as somatic mutations and chromosome amplification, which are associated with the tumors from which they originated[5]. These genetic markers are considered useful not only for predicting cancer but also for assessing the risk in other patients. Importantly, apoptosis and other forms of cell death are thought to be the primary mechanisms behind the release of cell-free DNA, underscoring the relevance of cfDNA analysis in clinical diagnostics[8,9].

More recently, cell-free DNAs have gained further attention since DNAs are shown to retain epigenetic information such as DNA methylation and chromatin architecture[8,10,11]. Epigenetic landscape is known to be tissue and cell type specific[12,13]. Therefore, this additional layer of information provides further opportunities to distinguish disease states from healthy conditions in cfDNA data. In fact, both DNA methylation and chromatin architecture have been demonstrated to be useful for detecting differential patterns present in various cancer-derived cfDNAs[5,10,14–16]. In terms of chromatin architecture, cfDNAs are shown to contain nucleosomes including histones with post-translational modifications[17–20]. Cancer patient derived cfDNAs contain unique nucleosome positioning or spacing, particularly at gene promoters, compared to those from healthy donors[5,21]. In addition to nucleosome footprints, transcription factors' footprints are also observed in short and subnucleosomal DNA fragments[5,22]. However, accurately distinguishing cancer-derived cfDNA from background cfDNA remains a significant challenge due to the complexity and variability of epigenetic patterns, as well as the typically low abundance of tumor-derived cfDNA[23–26]. Recently, machine learning approaches have demonstrated efficacy in cfDNA prediction analysis, showing promise in distinguishing cancer types, origins, and properties[5,27–29]. Integration of chromatin architecture, such as tissue-specific chromatin accessibility, together with tumor-relevant mutations, DNA methylation, and cfDNA occupancy, represents a

[1]Department of Biomedical Sciences, University of North Dakota School of Medicine and Health Sciences, Grand Forks, ND, USA. [2]Divisions of Surgical Oncology and Hepatobiliary Surgery, Los Angeles General Medical Center, Los Angeles, CA, USA. [3]Sanford Research, Sioux Falls, SD, USA. [4]Department of General Surgery, Graduate School of Medicine, Chiba University, Chiba, Japan. [5]Department of Neurology, University of Tennessee Health Science Center, Memphis, TN, USA. ✉e-mail: motoki.takaku@und.edu

promising direction for the clinical application of cfDNA as a diagnostic tool[30,31]. However, numerous challenges persist for translational use, encompassing not only prediction accuracy and generalizability but also data interpretability and the intricacy of data processing[29]. To address these challenges, interpretable machine learning methods hold significant potential to deliver both predictive performance and actionable biological insights. Several studies have shown that cfDNA enrichment patterns partially reflect ongoing biological processes within tumors[32–34]. Therefore, incorporating signals from non-tumor genes and elements of the tumor microenvironment, including immune cells and immune responses, may enhance our ability to detect cancer at early stages and monitor treatment responses, including those to immunotherapy.

In this study, we collect cfDNAs from the blood of breast cancer and pancreatic cancer patients. Consistent with prior research, typical nucleosome patterns were observed at gene promoters. Furthermore, both breast and pancreatic cancer cfDNAs exhibited enrichment at open chromatin regions. Notably, differential enrichment between healthy donors and cancer patients was observed at open chromatin regions often associated with breast or pancreatic cancer, as well as CD4$^+$ T cells. To further evaluate the significance of open chromatin regions in cfDNA analysis and address signal variations among patient cfDNA data, we applied a machine learning approach. The XGBoost machine learning approach demonstrated a distinct improvement in cancer patient prediction when using cell type-specific open chromatin features for breast and pancreatic cancer, but not for ovarian cancer. The trained model identified specific chromosomal regions that contributed significantly to prediction accuracy. These findings underscore the utility of cfDNA enrichment signals at open chromatin regions and highlight the potential of combining interpretable machine learning with biologically informed features to reveal cancer-specific chromatin landscapes preserved in cfDNA.

## Results

### Cell-free nucleosomal DNAs are enriched at open chromatin

To characterize the fundamental patterns of cfDNAs released from cancer cells, we first collected in vitro cfDNA from luminal breast cancer cell lines (T47D and KPL-1). Cells were cultured to confluence, after which the culture medium was collected and carefully filtered to remove cellular debris (see Methods). This in vitro approach enables the isolation of cfDNA fragments released directly from cancer cells, without the confounding influence of immune cell contributions or enzymatic DNA degradation that typically occurs in the bloodstream[35–38]. Isolated DNAs released from the cells showed clear nucleosomal DNA fragment patterns (Fig. 1a). It should be noted that the observed fragment sizes reflect the presence of sequencing adapters (~100 bp), which causes a shift in apparent fragment size in the Tapestation profile. These nucleosomal fragments are clearly enriched at open chromatin regions. When ATAC-seq peaks (51,463 peaks) in T47D cells are used as a reference for open chromatin regions, in vitro cfDNAs isolated from T47D cell culture medium are accumulated at the center of T47D ATAC-seq peaks, with a typical well-positioned nucleosomes flanked at the peak center, patterns frequently observed at active enhancers (Fig. 1b). The cfDNA signals from KPL-1 cells, another luminal breast cancer cell line, also showed clear enrichment at T47D ATAC-seq peaks (Supplementary Fig. 1a). To confirm if these patterns are conserved in patient derived cfDNAs, we isolated cfDNAs from five luminal breast cancer patients (Stage 0 or I) and six healthy donors using 600 μL of human plasma, after which next-generation sequencing libraries were prepared (Fig. 1a, Supplementary Fig. 1b, Supplementary Table 1). DNA fragment size analysis by Tapestation confirmed that patient cfDNAs again preserved typical chromatin DNA fragmentation patterns, with DNA fragments corresponding to mono-, di-, and tri-nucleosomes (Fig. 1a). The DNA fragment length was further confirmed by sequencing data (Fig. 1c). Consistent with previous studies[5,19], cfDNAs from both breast cancer patients and healthy donors showed striking enrichment ~167 bp, suggesting chromatosome (linker histone and nucleosome complex). In subnucleosomal fragments ranging from 80 to 140 bp, a periodicity of approximately 10–11 bp was also observed, further suggesting the presence of chromatin or nucleosome signatures (Fig. 1c)[39].

Similar to in vitro cfDNAs, genome browser tracks showed that patient derived nucleosomal fragments (cfNuc) were frequently enriched at gene promoters and open chromatin regions detected in T47D ATAC-seq data (Fig. 1d), while large sample variations and extra signals in other genomic loci are present and notable in human plasma derived cfNuc. Metagene plot analysis of transcription start sites (TSS) (13,961 genes) displayed the typical chromatin landscape at promoters, and well-positioned nucleosomes, known as +1 nucleosomes, were observed downstream of promoters (Fig. 1e). Metaplot analysis at T47D ATAC-seq peaks and breast cancer-specific enhancers confirmed higher enrichment of cfNuc at open chromatin regions (Fig. 1f, Supplementary Fig. 1c). While differential enrichment of cfNuc between breast cancer and healthy donors was observed in Genome Browser tracks, mean signals at open chromatin regions were similar between cancer patient derived and healthy donor derived samples (Fig. 1f). To assess whether the observed enrichment was influenced by the relatively low sequencing depth (~30 million reads; see Methods), we performed deep sequencing (100 million reads) on a subset of samples. This analysis confirmed that the enrichment at open chromatin regions is not merely a consequence of sequencing depth (Supplementary Fig. 1d). To further confirm the quality of our cfDNA purification and sequencing data, we examined the DNA fragment end motifs. Prior studies have shown that cfDNA fragmentation is non-random and often reflects the activity of endogenous nucleases. In particular, the CCNN motif has been associated with the cleavage preference of DNase I-like enzymes, such as DNASE1 and DNASE1L3, which are involved in apoptotic DNA fragmentation and circulating DNA clearance[40–43]. Consistent with these findings, our cfNuc fragments displayed a higher frequency of CCNN motifs at their 5′ ends (Supplementary Fig. 1e). However, those motifs were not specifically enriched at open chromatin regions (Supplementary Fig. 1f). Finally, we attempted to estimate the fraction of tumor derived cfDNAs using ichorCNA, a method based on copy number variation profiling. The analysis suggested that tumor fractions ranged from approximately 1% to 3% (Supplementary Fig. 1g). This relatively low proportion may reflect the clinical status of the patients, as all samples were collected at early stages of breast cancer, and most of them were obtained after surgery or treatment (Supplementary Table 1). In summary, our nucleosome fragments isolated from breast cancer patients retain typical chromatin landscapes at open chromatin regions.

### Distinct cfDNA signals in breast cancer patients are detected in breast cancer-specific open chromatin

Although the average signal intensities of cfNuc data in breast cancer patients and healthy donors were not significantly different, differential signals were evident from genome browser tracks, frequently associated with luminal breast cancer signature genes (Fig. 2a). When read counts were collected from gene promoters (1 kb bin size) associated with apoptosis, cell cycle, and cell stress response pathways, significant differences were observed in multiple genes, including FADD and CDK9 (Fig. 2b–d). Additionally, genes related to mammary gland development, such as FOXA1, TBX genes, and WNT genes, displayed significantly elevated cfNuc signals, suggesting cell type-specific enrichment of cfNuc in breast cancer samples (Fig. 2e).

To identify differentially enriched nucleosome fragments across the genome, we conducted EdgeR differential peak analysis on T47D ATAC-seq peaks (51,463 genomic loci). When an unadjusted p-value threshold of <0.05 was applied, we identified 776 peaks with increased cfNuc occupancy and 871 peaks with decreased occupancy in breast cancer samples compared to healthy donors (Fig. 2f). Among these, 38.3% of the differentially enriched regions were found near promoters (Fig. 2g), slightly increased but similar to the distribution of the T47D ATAC-seq peaks used as a reference (32.8% for T47D ATAC-seq) (Supplementary Fig. 2a). To further dissect the biological pathways associated with these regions, we assigned differential peaks to the

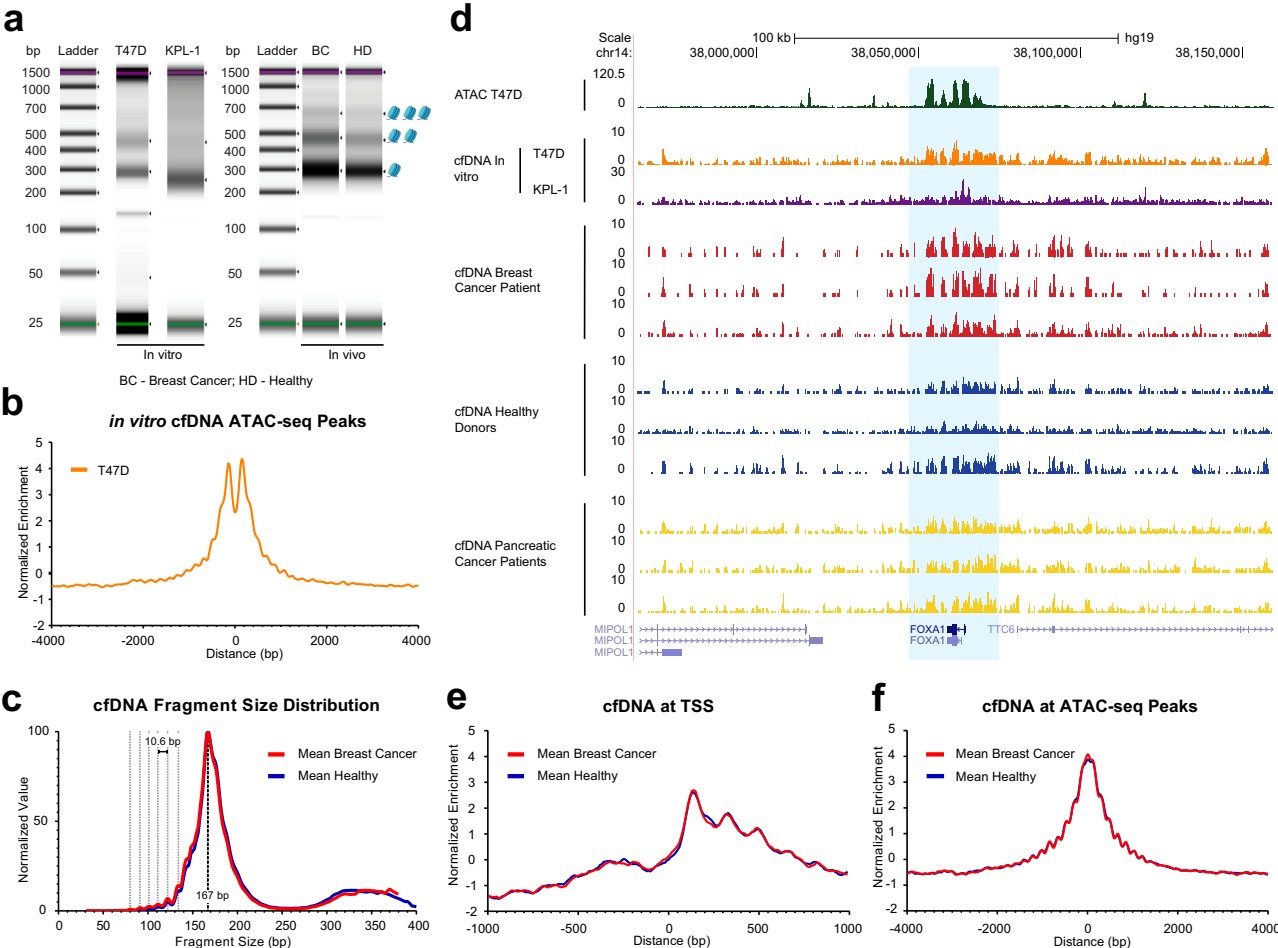

**Fig. 1 | Enrichment of cfDNAs at open chromatin regions. a** Tapestation data shows the DNA fragment size distributions of in vitro (cell culture isolated) and in vivo (patient derived) cfDNA sequencing libraries. T47D and KPL-1 luminal breast cancer cells were cultured to confluence, and cfDNAs were purified from the culture medium. Plasma from breast cancer (BC) patients and healthy donors (HD) was used to purify in vivo cfDNA. Note that the observed size distributions include sequencing adapter sequences ( ~ 100 bp total), which result in a size shift in the Tapestation profiles compared to the actual cfDNA fragment sizes. **b** Metaplot of in vitro cfDNA enrichment displaying cfDNA enrichment at T47D ATAC-seq peaks. The cfDNAs were purified from T47D cell culture medium. **c** Fragment size distribution of cfDNA. Mean fragment sizes from each group were normalized and scaled from 0 to 100 for comparison and plotted. **d** Genome browser tracks with representative images showing cfDNA coverage. ATAC-seq data from T47D cells serve as the reference for open chromatin. Coverage signals from both in vitro and in vivo cfDNAs are displayed, including three biological replicates of patient-derived cfDNA. **e** Patient-Derived cfDNA enrichment at TSS illustrates fragment enrichment patterns from breast cancer patients and healthy donors at transcription start sites (TSS). **f** Metaplot shows in vivo cfDNAs normalized mean enrichment at T47D ATAC-seq peaks. Normalized mean reads were calculated from cfDNA data of breast cancer patients and healthy donors.

closest genes. Subsequent pathway enrichment analysis confirmed that pathways related to luminal breast cancer, such as breast cancer, estrogen signaling, and endocrine resistance, as well as cfDNA biology-related pathways, including apoptosis and cellular senescence, are significantly associated with differentially enriched cfNuc regions (Fig. 2h, Supplementary Fig. 2b). Additionally, cancer-relevant signaling pathways, such as Hippo and Wnt signaling, were also detected among the top enriched pathways. These results suggest that differentially enriched cfNuc regions are associated with their tumor origin and the biology of cell-free DNA release.

## Differentially enriched cfDNAs are associated with biological pathways related to cancer and immune cell functions

Given that the majority of cfDNAs are known to be derived from immune cells, we sought to identify differentially enriched loci associated with open chromatin regions in immune cells. In the tumor microenvironment, CD4+ T cells have been reported to accumulate around breast tumors in vivo[44,45]. Our FACS analysis of PBMC fractions from breast cancer patients confirmed a higher frequency of CD4+ T cells, although the proportions varied among individuals (Supplementary Fig. 3). Based on this, we used ATAC-

seq peaks identified in human CD4+ T cells (42,517 genomic loci)[46] as a reference for differential peak analysis. Using a p-value threshold of <0.05, 567 peaks showed increased signals, and 649 peaks exhibited decreased signals (Fig. 3a). Peak annotation analysis indicated that 39.6% of the differential peaks were associated with promoters (Fig. 3b, Supplementary Fig. 4a), similar to the fraction of differential loci found in breast cancer ATAC-seq based analysis, as shown in Fig. 2g. KEGG and GO pathway analysis detected T-cell-related pathways, while other pathways, such as breast cancer, Wnt signaling, and cellular senescence, are commonly found in both T47D and CD4+ T cell ATAC-seq peak analyses (Fig. 3c, Supplementary Fig. 4b). Additionally, we used EnhancerAtlas 2.0 to further collect breast cancer-specific enhancers and promoters and obtained 1986 unique differential peaks linked to T47D cell open chromatin. When comparing differentially enriched regions between T47D and CD4+ T cells, 398 peaks (out of 1986 peaks, ~20%) overlapped (Fig. 3d, Supplementary Fig. 4c). To determine if these differential peaks sufficiently distinguish between samples from breast cancer patients and healthy donors, we compiled a peak list containing all differentially enriched regions from both T47D and CD4+ T cells (2804 genomic regions, Supplementary Data 1). The MDS plot clearly differentiated the samples from healthy individuals and breast cancer

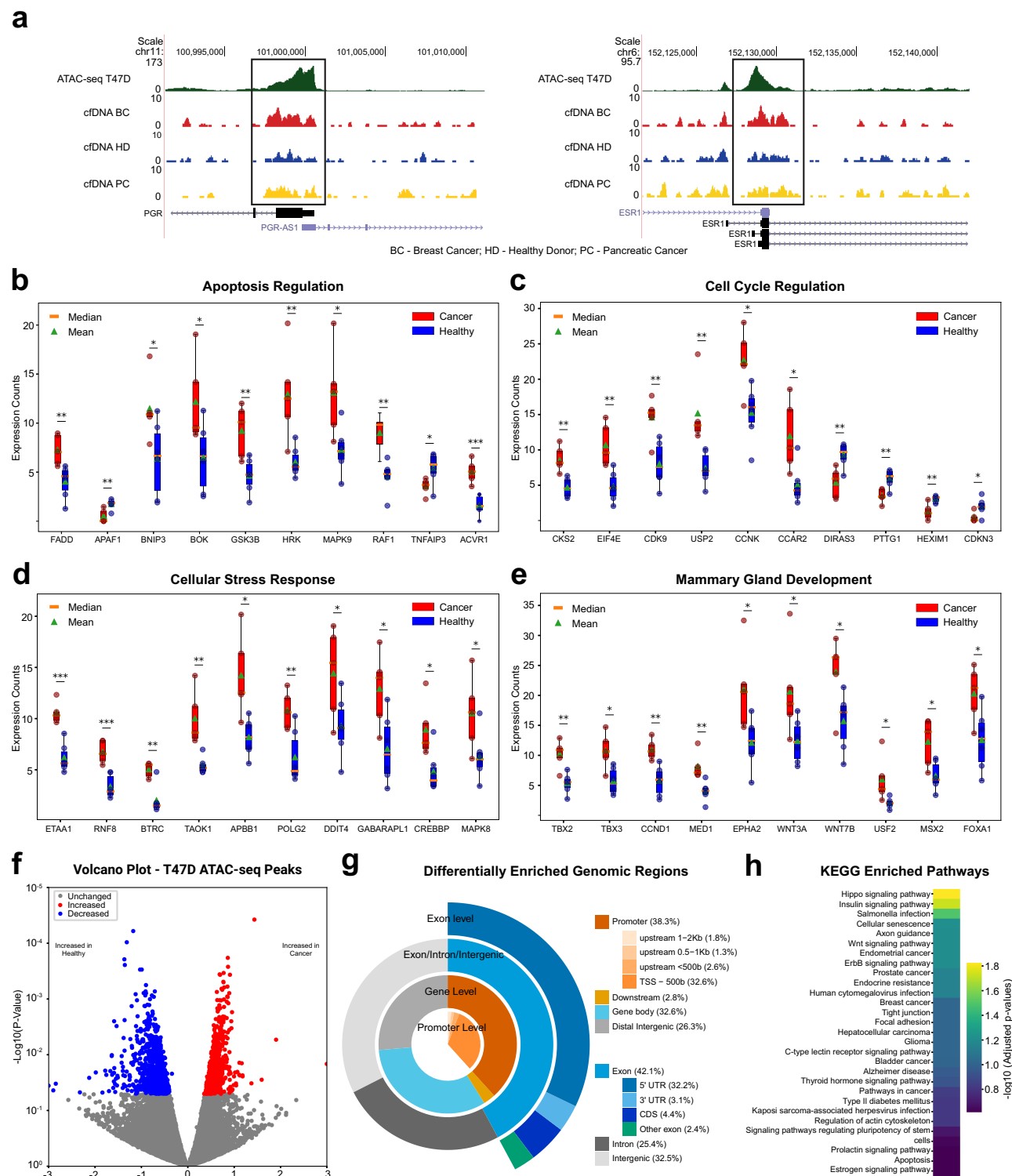

Fig. 2 | Differential enrichment of cfDNAs in breast cancer versus healthy samples at luminal breast cancer open chromatin regions. a Genome browser tracks of cfNuc signals at luminal marker gene promoters. ATAC-seq data from T47D cells serve as the reference for open chromatin. Genome coverage of cfNuc from breast cancer (BC), healthy donor (HD), and pancreatic cancer (PC) samples are displayed. b Box plots showing normalized cfNuc read counts at apoptosis related gene promoters in breast cancer ($n = 5$, red) and healthy donor ($n = 6$, blue). Statistical significance was assessed by unpaired t-test: p < 0.05 (*), p < 0.01 (**), p < 0.001 (***). Whiskers denote values within 1.5x the interquartile range; values outside this range are visualized as individual points. c–e Box plots showing normalized cfNuc read counts at gene promoters related to cell cycle regulation (c)

cellular stress response (d) and mammary gland development (e). f Differential enrichment analysis at T47D ATAC-seq peaks. The volcano plot illustrates differential signals between breast cancer and healthy donor cfDNAs. Regions with significantly increased or decreased cfNuc occupancy (p < 0.05) are highlighted in red and blue, respectively. g Peak annotation analysis of differentially enriched regions. The circular chart shows the distribution of differentially enriched genomic regions, categorized by promoter, exon, intron, and intergenic areas, indicating their frequency. h KEGG pathways enrichment analysis. Differentially enriched regions are associated with the closest genes, and enriched pathways are displayed using a gradient scale representing the $-\log_{10}$ of adjusted p-values.

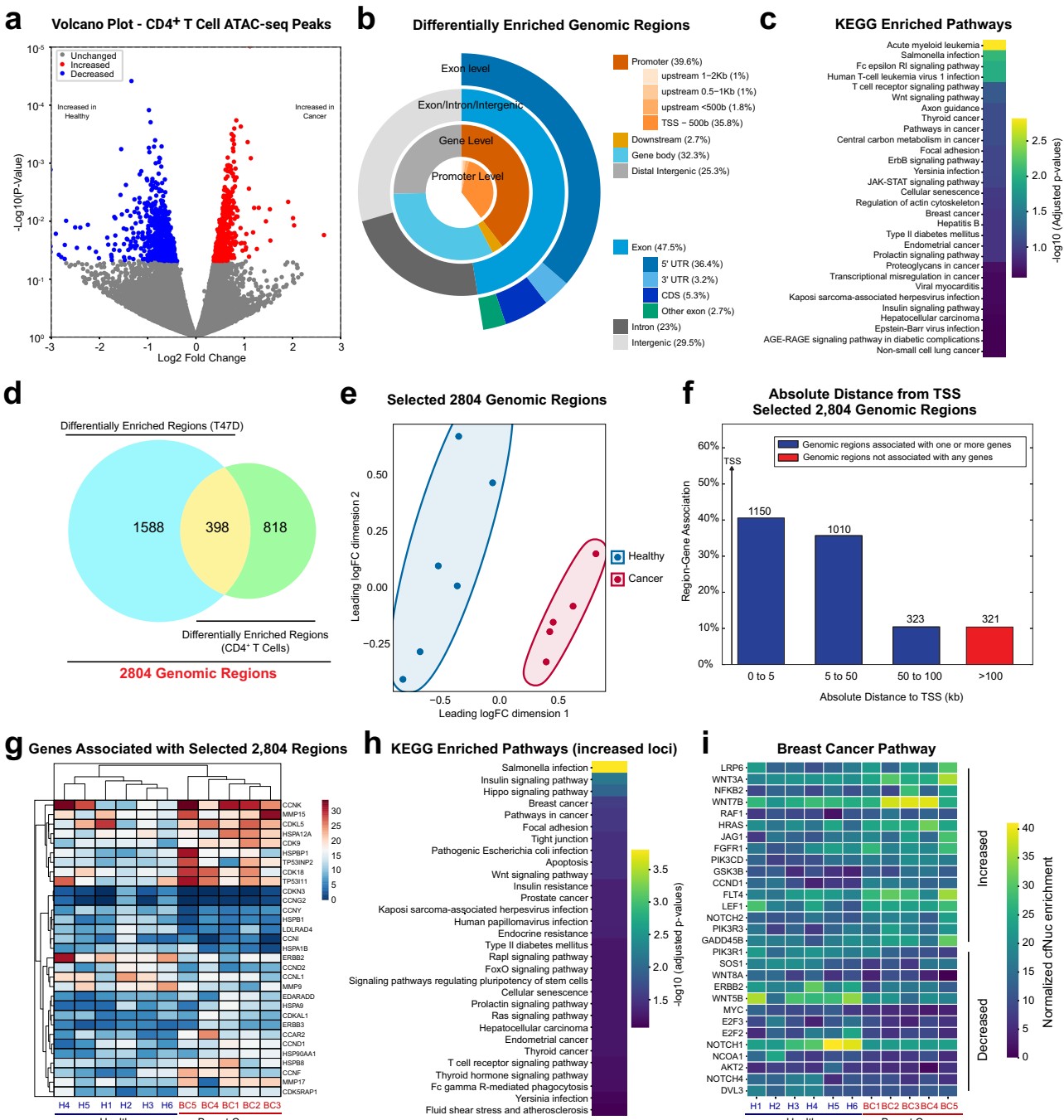

**Fig. 3 | Integrative analysis of cfNuc enrichment using CD4+ T cell and breast cancer ATAC-seq peaks. a** Differential enrichment analysis at CD4+ T cell ATAC-seq peaks. The volcano plot shows differential signals between breast cancer and healthy donor cfDNAs. Regions significantly (p-values < 0.05) increased and decreased are marked in red and blue, respectively. **b** Peak annotation analysis of differentially enriched regions. Differentially enriched loci are categorized into promoter, exon, intron, or intergenic regions. **c** KEGG pathway enrichment analysis of genes associated with differential loci based on CD4+ T cell ATAC-seq peak analysis. **d** Overlap of differentially enriched regions. Venn diagram depicts the intersection of differentially enriched regions identified by T47D or CD4+ T cell ATAC-seq peak based analysis. A total of 2804 regions is contained in the merged list (Supplementary Data 1). **e** Multidimensional scaling (MDS) plot demonstrating

distinct clustering of cfNuc signals between healthy and breast cancer samples, based on 2804 selected genomic regions. **f** Distribution of 2804 merged regions relative to TSS. **g** Heatmap displaying normalized cfDNA read counts for genes associated with the top differentially enriched regions. Read counts were derived from differential peaks and mapped to their nearest genes. Color intensity reflects relative read count values. **h** KEGG pathway enrichment analysis of genes associated with regions of increased cfNuc occupancy. Pathway analysis was performed using genes linked to genomic regions showing higher cfNuc enrichment in breast cancer samples.
**i** Heatmap displaying normalized cfNuc read counts for genes associated with the KEGG breast cancer pathway. Each column represents an individual sample, and each row corresponds to a gene. Color intensity reflects the relative level of cfNuc enrichment at genomic regions linked to each gene.

patients (Fig. 3e). Since many loci are proximal to TSS (Fig. 3f, Supplementary Fig. 4d), the 2804 selected peaks were associated with genes, and gene ontology analyses were performed. As expected, genes involved in both breast cancer and immune cell pathways were significantly enriched (FDR < 0.05) (Fig. 3g–i, Supplementary Fig. 4e). These results demonstrate that ATAC-seq-guided comparative analysis effectively identifies enriched chromatin signatures in breast cancer cfDNA data, and importantly, reveal that differential cfDNA enrichment is not limited to cancer cell specific open chromatin regions but also occurs at immune cell specific regions, such as those in CD4$^+$ T cells. This suggests that cfDNA profiles may reflect both tumour-intrinsic and immune microenvironment-derived chromatin landscapes.

### Cancer type-specific enrichment is detected in pancreatic cancer and breast cancer comparison

To assess whether differentially enriched cfNuc regions identified in breast cancer patients are specific to breast cancer or shared across other cancer types, we performed a similar analysis using cfNuc from pancreatic cancer patients. Plasma cfDNA was purified from pancreatic cancer patient samples ($n = 8$, Supplementary Table 1), and the resulting DNA showed nucleosomal fragment patterns, with a dominant peak corresponding to mono-nucleosomes (Supplementary Fig. 5a). Genome browser tracks revealed that cfNuc occupancy in pancreatic cancer samples displayed distinct enrichment patterns compared to those observed in breast cancer patients and healthy donors, suggesting cancer type-specific chromatin signatures (Fig. 4a, Supplementary Fig. 5b). As observed in breast cancer and healthy donor samples, cfNuc signal intensities in pancreatic cancer patients were highest at the centers of pancreatic cancer-specific open chromatin regions (3571 loci) obtained from EnhancerAtlas 2.0 (Fig. 4b). From EnhancerAtlas 2.0, we extracted 3185 promoter and 386 enhancer regions specific to pancreatic cancer. Differential enrichment analysis revealed a modest number of increased (176 peaks) and decreased (141 peaks) regions in pancreatic cancer cfNuc compared to healthy controls (Fig. 4c), with a slightly higher proportion of peaks in gene body and intergenic regions (Supplementary Fig. 5c). Despite the relatively small number of differentially enriched regions, they were sufficient to distinguish cfNuc profiles of pancreatic cancer patients from healthy donors (Fig. 4d).

When CD4$^+$ T cell ATAC-seq peaks were used as the reference, a larger set of differentially enriched loci was detected, showing a consistent trend toward increased cfDNA signals in pancreatic cancer samples (Fig. 4e, Supplementary Fig. 5d). Gene ontology analysis confirmed that the differentially enriched regions from both pancreatic cancer-specific and CD4$^+$ T cell open chromatin were significantly associated with pancreatic cancer related pathways (Fig. 4f–g, Supplementary Fig. 5e, f). Importantly, a merged set of differential peaks derived from breast cancer cfNuc analysis successfully differentiated cfDNA profiles between breast cancer and pancreatic cancer patients (Fig. 4h). These results demonstrate that ATAC-seq peak guided analysis effectively captures cancer type-specific cfNuc enrichment patterns and can distinguish between different tumor types based on chromatin features preserved in cfDNA.

### Machine learning analysis guided by ATAC-seq enhances prediction of cancer cfDNA

To evaluate the findings from our cfDNA sequencing data, cfDNA data from 64 breast cancer patients and 57 healthy donors were collected from a public cfDNA data bank[47]. Considering the substantial variability in human cfDNA specimens, we opted for machine learning analysis (Supplementary Fig. 6a). XGBoost, known for its effectiveness in handling partially conserved features[48], was used to predict breast cancer cfDNA patterns. We divided our dataset into a 70% training set and a 30% test set to assess the prediction accuracy of our models. When 2804 breast cancer cfNuc differential peaks were used for the XGBoost prediction analysis, it achieved a relatively high prediction accuracy of 85.29% (AUC 92%, validation 92.65%, cross-validation 84.34%) compared to randomly selected genomic regions (Fig. 5a, Supplementary Fig. 6b. Supplementary Table 2), suggesting that the

selected loci were able to detect unique chromatin features conserved in a different data cohort (Supplementary Fig. 6c, d).

Although the cfDNA signals were enriched at promoters and open chromatin, signals can be seen in other chromatin loci. To further comprehensively investigate the enriched cfDNA signature, we repeated the same prediction analysis using the genome coverage data from the entire genome (Fig. 5b). Relatively high prediction scores were observed when chromosome regions were segmented using a 10 kb bin size. Interestingly, even higher overall prediction accuracy was detected when the training data were confined to open chromatin regions, particularly those using combined ATAC-seq peaks from T47D and CD4$^+$ T cells.

We extended this approach to cfDNA datasets from pancreatic and ovarian cancer patients, analyzing previously published data comprising 34 pancreatic cancer samples and 27 ovarian cancer samples[15]. In the pancreatic cancer dataset, the highest prediction accuracy was achieved when using pancreatic cancer-specific enhancer and promoter regions (3,571 loci) as input features (Supplementary Fig. 6e, Supplementary Table 2). The inclusion of CD4$^+$ T cell specific open chromatin regions did not result in a substantial improvement in prediction scores. In contrast, for ovarian cancer cfDNA, the best prediction performance was obtained using genome-wide coverage data. Incorporating ovarian cancer ATAC-seq peaks derived from PEO1 cell line[49] or CD4$^+$ T cell ATAC-seq peaks did not enhance model accuracy and, in some instances, slightly reduced it (Supplementary Fig. 6e, Supplementary Table 2). These findings suggest that chromatin-based cfDNA signatures may be cancer type-specific and that optimal predictive features may differ across cancer types. Taken together, the results from the machine learning prediction analysis underscore the efficacy of using cell type-specific open chromatin regions to identify unique patterns in cancer patient cfDNA data.

### Interpretable machine learning model identifies novel biomarkers for breast cancer detection

XGBoost is a powerful, decision tree-based ensemble algorithm known for its efficiency and interpretability. The trained models provide decision trees and rank important features, which are easily interpretable (Fig. 5c). From the prediction analysis using 2804 breast cancer cfNuc differentially enriched regions, 52 top signatures were extracted from the trained model, and these regions were sufficient to separate cfNuc patterns between breast cancer and healthy donor samples in our data (Supplementary Fig. 6f). Significant features identified by the trained model 4 (Fig. 5b) include chromosome regions associated with genes known to influence tumor development (Fig. 5d, Supplementary Fig. 6g). Among these, EMP3 was found as the top signature-associated gene, yet its roles in tumor biology are poorly reported. Kaplan-Meier patient survival analysis in breast cancer cases revealed that EMP3 expression levels are positively correlated with patient survival (Fig. 5e). Additionally, co-expression analysis in the METABRIC breast cancer cohort[50] found that S100A4 exhibited the strongest correlation (Fig. 5f). The involvement of S100A4 is well-documented across various tumors, including breast tumors. These results demonstrate the advantages of using ATAC-seq guided machine learning analysis for data interpretation.

## Discussion

Chromatin structure outside nuclei or cells has increasingly become a target for both basic biology and translational studies. Nucleosome footprints and DNA methylation patterns in cell-free DNA are valuable tools for detecting DNA fractions derived from diseased tissues. Multiple studies have confirmed the presence of nucleosomes and histones in plasma and serum using techniques such as mass spectrometry, chromatin immunoprecipitation, and other methods[17,20,51]. Additionally, these histones in nucleosomes retain post-translational modifications[17,52,53], providing further evidence that cell-free nucleosomes can partially reflect the gene expression of their cell of origin. However, because both active and suppressive histone modifications are detected in plasma chromatin, the full implications of cell-free

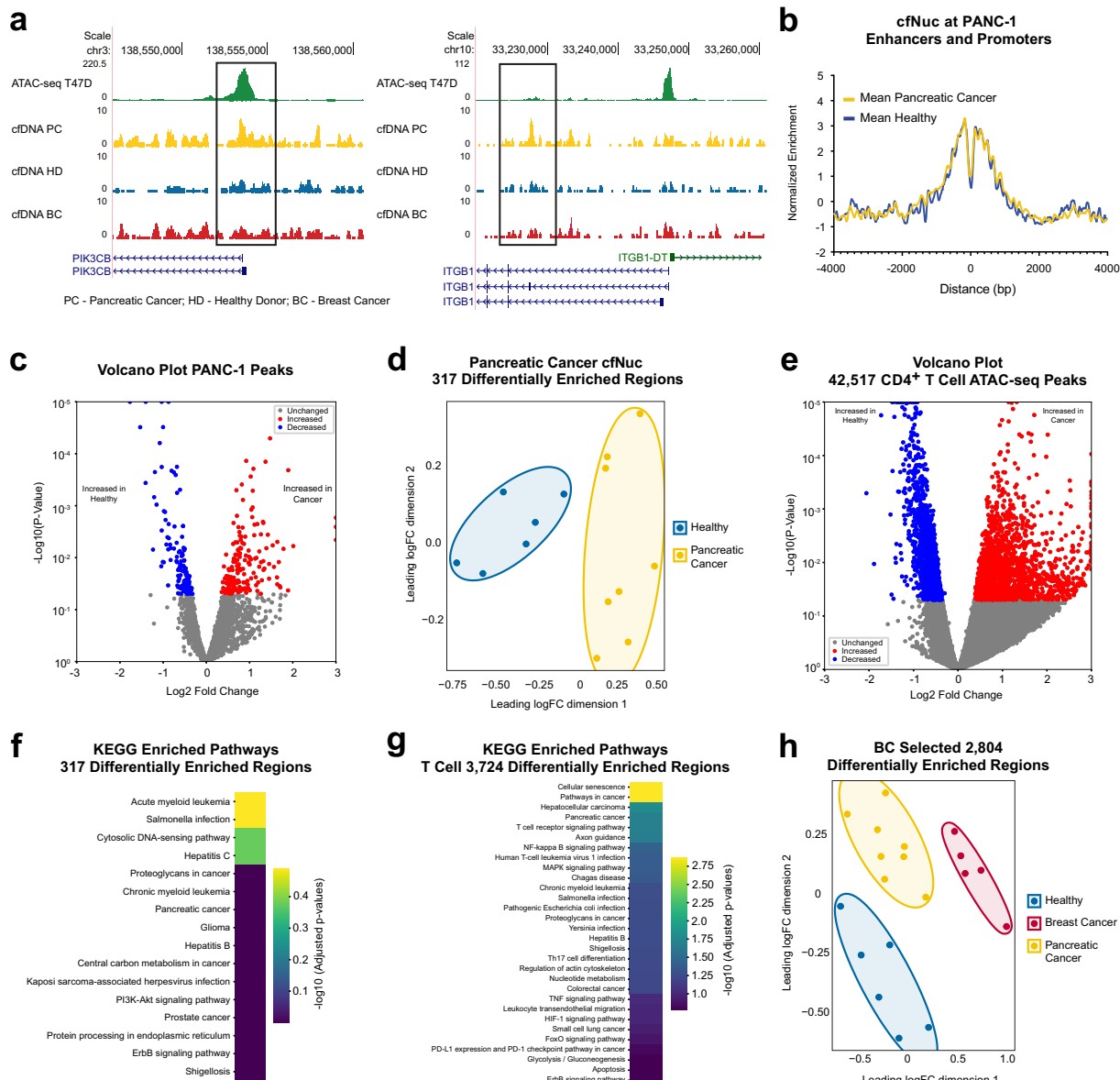

**Fig. 4 | Differential enrichment analysis of cfDNAs in pancreatic cancer.**
**a** Genome browser tracks of cfNuc signals at pancreatic cancer related gene promoters. Genome coverages are extracted from pancreatic cancer (PC), healthy donor (HD), and breast cancer (BC) cfDNA data. ATAC-seq T47D cells is shown as a reference for open chromatin regions in breast cancer cells. **b** Metaplot comparing cfDNA enrichment at PANC-1 enhancer and promoter regions. Mean enrichment profiles of cfDNA from pancreatic cancer patients and healthy controls are plotted. **c** Volcano plot for differential peak analysis using cfDNAs isolated from pancreatic cancer. Differentially enriched regions were identified with a p-value threshold of <0.05. **d** MDS plot showing clustering of cfDNA profiles from healthy individuals and pancreatic cancer patients based on signals at differential peaks. **e** Volcano plot

for differential enrichment analysis at CD4+ T cell ATAC-seq peaks. Increased and decreased regions were identified with a p-value threshold of <0.05. **f** KEGG pathway enrichment analysis using genes associated with differential peaks. Pathway analysis was performed on 317 differentially enriched loci from pancreatic cancer-specific open chromatin regions. **g** Heatmap showing significantly enriched pathways. KEGG pathway analysis was conducted on differential loci identified at T cell ATAC-seq peaks. **h** MDS plot of cfDNA profiles from healthy, breast, and pancreatic cancer samples. Distinct clustering based on 2804 selected differential peaks demonstrates clear separation by sample type. The merged peak list created in Fig. 3 was used for analysis.

nucleosomes, along with these modifications, have yet to be fully characterized and warrant further investigation through fragmentomics studies.

In this study, we utilized open chromatin information determined by ATAC-seq in relevant cancer and immune cells in vitro to dissect chromatin signatures conserved in plasma from patients with luminal breast cancer and pancreatic cancer. Experimentally defined ATAC-seq peaks in the T47D luminal breast cancer cell line proved useful for identifying differentially enriched genomic regions in breast cancer cfDNAs or cfNuc data. The efficacy of using ATAC-seq peaks was further validated by cfDNA data

from pancreatic cancer patients. Additionally, differential signals at ATAC-seq peaks derived from T cells were observed in breast and pancreatic cancer cfDNA datasets, suggesting alterations in cell-free nucleosomes from immune cells and/or changes in the ratios between DNA fragments derived from immune and cancer cells in patient blood. Gene ontology analysis revealed that differentially enriched nucleosomes are associated with cfDNA biology, such as apoptosis, as well as cancer type specific pathways. In the case of breast cancer–derived cfNuc, pathways related to estrogen signaling, endocrine resistance, and mammary gland epithelial cell proliferation were significantly enriched in breast cancer samples compared to healthy donors

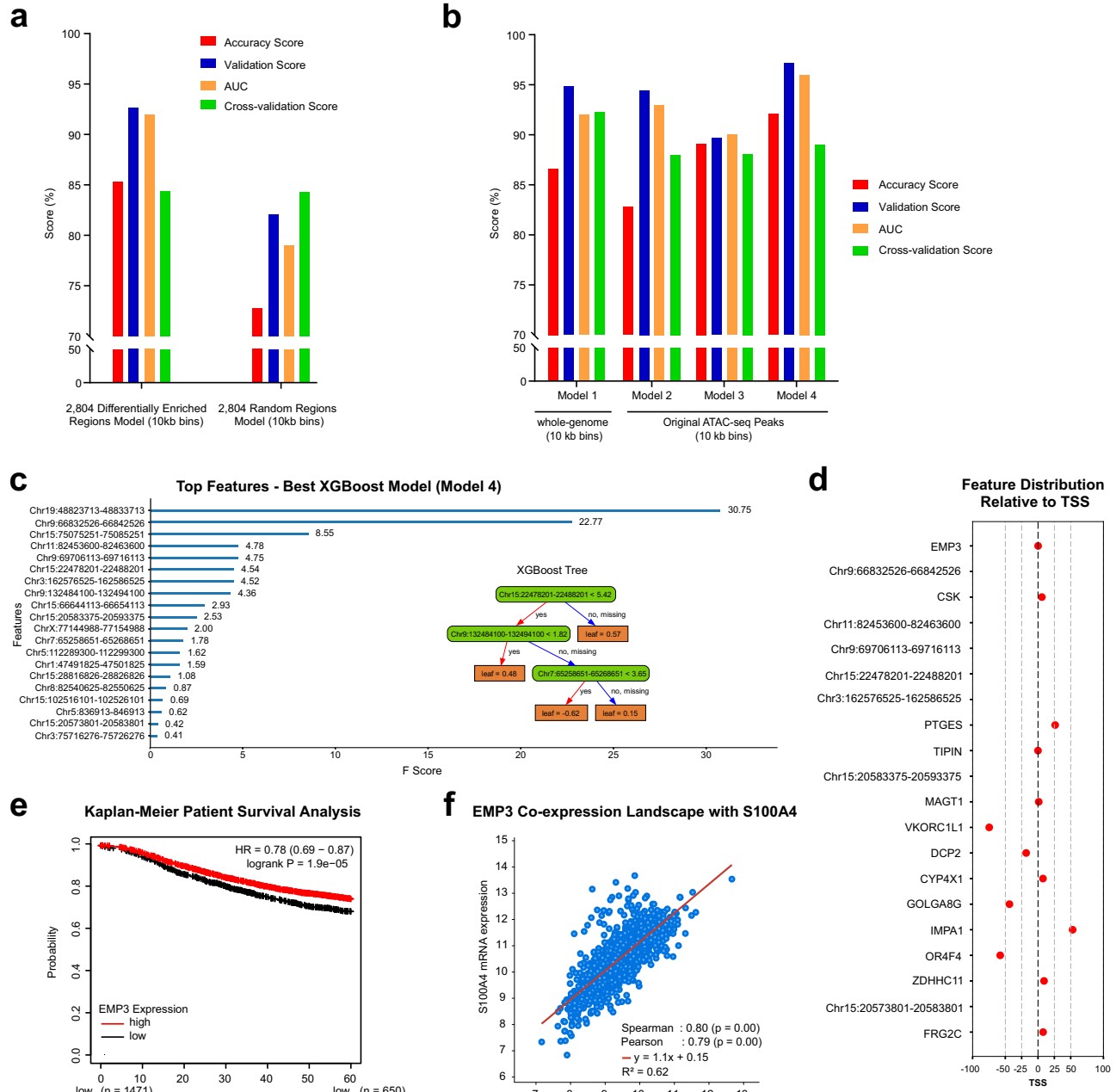

**Fig. 5 | Application of XGBoost machine learning to identify cancer-specific patterns in cfDNA. a** Model performance with 2804 selected open chromatin loci. Bar graphs indicate the accuracy, validation scores, and AUC for models using breast cancer differential peaks versus randomly selected loci. Each score is defined in the Methods section. To better accommodate potential impacts of chromosome amplification, each locus was expanded to 10 kb. **b** Performance comparison across multiple models. Genome coverage data from the whole-genome (in 10 kb bins) and 10 kb expanded ATAC-seq peaks were used for XGBoost modeling. Model 1 utilizes whole-genome coverage at 10 kb resolution; Model 2 uses 10 kb expanded peaks from T47D ATAC-seq; Model 3 employs 10 kb expanded peaks from CD4+ T cell ATAC-seq; Model 4 combines the 10 kb expanded peaks from both T47D and CD4+ T cell ATAC-seq. **c** Top significant features extracted from the best XGBoost model (Model 4). An example of XGBoost decision trees is also shown. **d** Distribution of significant features relative to TSS. **e** Kaplan–Meier patient survival analysis based on EMP3 expression. **f** Scatter plot showing the correlation of EMP3 and S100A4 expression levels in breast cancer. METABRIC[50] cohort was used.

(Fig. 2). Due to the limited number of our patient samples, we expanded our investigation to publicly available cfDNA data[47]. To address signal variability across patient samples, we applied the XGBoost machine learning model due to its proven scalability, ability to capture complex and partially conserved patterns, and its interpretability[48,54–57]. By training the model with differential ATAC-seq peaks identified from our small dataset, we achieved relatively high prediction accuracy (accuracy 85.29%, AUC 92%, validation 92.65%, cross-validation 84.34%). This accuracy further improved upon integrating ATAC-seq peaks from both breast cancer and immune cells

(accuracy 92.06%, AUC 96%, validation 97.14%, cross-validation 89.04%). The XGBoost model's unique features allowed us to easily identify critical chromatin regions for detecting patient-specific fragment enrichment patterns. Additionally, we identified a gene not previously associated with breast cancer, whose expression levels significantly correlate with patient survival. Similar to the breast cancer prediction analysis, XGBoost prediction for pancreatic cancer performed best when the model was trained using cfDNA enrichment levels at pancreatic cancer-specific open chromatin regions (enhancers and promoters). The inclusion of cfDNA signals at

CD4$^+$ T cell specific open chromatin regions did not improve prediction accuracy in the pancreatic cancer dataset. Interestingly, in the case of ovarian cancer, neither ovarian cancer-specific nor T cell specific open chromatin regions yielded strong predictive performance. Instead, the highest accuracy was achieved when the model was trained using cfDNA signals across the entire genome. These results suggest that the predictive power of chromatin features varies by cancer type and that whole-genome cfDNA coverage may be more informative for certain cancers, where tumor specific chromatin signatures may be less pronounced or more heterogeneous. However, it is important to note that all machine learning analyses in this study were conducted on relatively small sample sizes, and further validation in larger, independent cohorts will be critical to confirm these findings. Additionally, the selection of genomic loci used for training substantially influences model performance. For instance, while approximately 50,000 ATAC-seq peaks were used in the breast cancer analysis, the pancreatic cancer model relied on only ~3000 loci yet still achieved relatively high prediction accuracy. For the ovarian cancer prediction analysis, we used ATAC-seq data from PEO1, a high-grade serous ovarian cancer (HGSOC) cell line. However, it is possible that this cell line does not fully capture the chromatin landscape of the ovarian cancer cohort used in our cfDNA dataset, which may have contributed to the lower prediction performance.

In addition to tumor intrinsic chromatin signatures, our data revealed differential cfDNA enrichment at CD4$^+$ T cell specific open chromatin regions in both breast and pancreatic cancer patients. This suggests that cfDNA fragmentation patterns may also reflect changes in the tumor microenvironment, particularly immune cell composition or activity. As the presence and dynamics of tumor infiltrating lymphocytes are known to correlate with treatment response, including to immunotherapies, cfDNA based chromatin profiling could provide a non-invasive strategy for monitoring immune activation or suppression in cancer patients. Future studies could explore whether dynamic changes in cfDNA signals at immune specific open chromatin regions are predictive of immunotherapeutic efficacy or immune related adverse events.

To fully leverage such signals and interpret the results, it is essential to understand the biological and technical factors that shape cfDNA fragment distribution across the genome. The exact molecular mechanisms underlying the higher enrichment of cfDNA or cfNuc at specific genomic regions remain unclear. Potential contributing factors may include preferential cleavage by nucleases at accessible chromatin sites, differential protection by nucleosomes, and cell type specific patterns of apoptosis or chromatin remodeling. It is also possible that some of the observed bias arises from technical or systematic influences. For instance, open chromatin regions may be more readily degraded and solubilized in plasma, resulting in overrepresentation during cfDNA isolation and library preparation. In contrast, heterochromatin and other closed chromatin regions may be less efficiently recovered, potentially leading to underrepresentation in sequencing data.

Nucleosome spacing analysis and other sophisticated methods have been utilized to detect cfDNAs from specific cancer type. However, data interpretation and the extraction of chromatin signatures often necessitate specialized task settings and are not readily accessible. Our established pipeline is relatively simple and can facilitate the creation of a gene (or chromatin) panel for translational studies, such as cfDNA-based diagnostic tools. Further analysis with a larger patient dataset is essential to fully assess the potential of ATAC-seq guided cfDNA analysis. Given that differential open chromatin regions correlate with gene expression, this method could provide an indirect yet effective means to explore the impact of cancer drugs on patients. Such comparative analysis of gene expression or chromatin accessibility in human biopsies often presents challenges, making this alternative approach potentially advantageous for patients.

In this study, we focused solely on nucleosomal fragment enrichment. However, integrating additional epigenetic and genetic information, such as transcription factor footprints, DNA methylation, histone modifications, and somatic mutations, alongside cfDNA enrichment data could further enhance our predictive capabilities[30–32,58]. Moreover, incorporating patient demographic and clinical information, including tumor stage and treatment history, may improve model performance. Notably, our breast cancer samples were predominantly from early-stage patients (Stage 0 or I), and many were collected after surgery or treatment. These clinical factors likely contributed to the relatively low tumor-derived cfDNA fractions estimated by copy number variation analysis. Despite this limitation, our open chromatin–guided approach successfully identified differential cfDNA enrichment patterns, highlighting its potential sensitivity in detecting subtle chromatin signatures associated with cancer. Further research is needed to understand whether combining these multi-layered features with cfDNA or nucleosome fragmentation profiles can refine cancer subtype classification, assess prognosis, and guide therapeutic decision-making.

## Methods

### Sample and data collection
Human blood plasma samples were obtained from healthy donors ($n = 6$), breast cancer patients ($n = 5$), and pancreatic cancer patients ($n = 8$) from Sanford Health Hospital Biobank. Only the luminal subtypes of breast cancer samples were used in this study. Additionally, samples from Pancreatic Ductal Adenocarcinoma (PDAC) patients were selected for pancreatic cancer analysis. Blood was collected in EDTA-coated lavender top tubes and centrifuged at $2500 \times g$ for 10 minutes at 4 °C. Plasma was carefully isolated without disturbing the buffy coat layer and stored at −80 °C until cfDNA extraction. The sample information is summarized in Supplementary Table 1.

Human blood plasma samples for Fluorescence-activated cell sorting (FACS) were obtained from luminal subtype breast cancer patients from Chiba University Hospital. Chiba University Hospital Review Board on Human Subjects approved this study (HS202201-02). This study complied with all relevant ethical regulations regarding patient data, in line with ethical norms and standards in the Declaration of Helsinki.

### Cell culture and in vitro cfDNA collection
T47D and KPL-1 breast cancer cell lines were cultured in high-glucose Dulbecco's Modified Eagle Medium (DMEM; Thermo Fisher Scientific) supplemented with 10% fetal bovine serum (Premium Select FBS; GeminiBio). Cells were maintained in 10 cm culture dishes and grown to confluence. The culture supernatant was then carefully collected and filtered through a 0.2 μm membrane to remove cellular debris. Filtered medium was stored at –80°C until cfDNA extraction. The T47D cell line was originally purchased from ATCC. The KPL-1 cell line was kindly provided by Dr. John Colicelli. Although KPL-1 is considered a clonal derivative of the MCF-7 breast cancer cell line, studies have reported phenotypic differences. In our study, KPL-1 was selected for in vitro cfDNA analysis due to its robust growth characteristics and more aggressive phenotypes compared to MCF-7. All cell lines were regularly tested and confirmed to be free of mycoplasma contamination.

### cfDNA isolation, library preparation, and sequencing
Blood plasma (600 μL) from cancer patients and healthy donors was used for cfDNA purification, following the manufacturer's protocol of the SubX$^{TM}$-Exo-DNA kit (Capital BioSciences). The kit separates exosome fraction first followed by cfDNA purification using magnetic beads. The purified cfDNA was quantified using the Qubit dsDNA HS Assay kit. The NEXTFLEX rapid DNA-seq 2.0 (Revvity) kit was employed for library preparation, using 30 μL of eluted cfDNA solution (ranging from 1 ng to 4 ng of total cfDNA) as input. Isolated cfDNAs underwent end-repair, adenylation, adapter ligation, and PCR amplification (9–11 PCR cycles). The amplified cfDNA was quantified again, and its quality was assessed using Tapestation (Agilent 4200) to analyze fragment size distribution. whole-genome sequencing was performed on the Illumina platform at Yale Center for Genome Analysis, achieving approximately thirty million reads per sample.

## Data processing and primary data analysis

Raw sequence reads were obtained from Yale Genomics Core in FASTQ format. Data pre-processing included initial read quality control and trimming to remove adapter sequences, eliminate low-quality reads, and trim low-quality base pairs. FastQC was used to assess data quality. Reads were then aligned to Homo sapiens genome assembly GRCh37 (hg19) using the Bowtie 1.2.2 splice-unaware aligner, with genomic sequences and annotation data provided via GTF/GFF and FASTA files[59]. Duplicate reads were eliminated using the MarkDuplicates tool from the Picard-tools-2.26.4 package (http://broadinstitute.github.io/picard/). Paired-end reads were converted into single fragments for metaplot analyses and visualization on the UCSC Genome Browser. Genome coverage was normalized to a fixed read depth of 10 million fragments. Previously published ATAC-seq data from the T47D breast cancer cell line was used as open chromatin reference[60]. Additionally, Gene Expression Omnibus was used to obtain open chromatin loci for CD4[+] T cells[46]. EnhancerAtlas 2.0 was used to obtain known promoter and enhancer peak coordinates for Pancreatic cancer (PANC-1)[61].

## Differential peak analysis and pathway enrichment analysis

To identify differential cfDNA signals, nucleosomal DNA fragments were collected at ATAC-seq peaks and compared between healthy donors and breast cancer patients or pancreatic cancer patients. All ATAC-seq peaks were extended to 1 kbp. Differentially enriched genomic loci were identified using EdgeR, with a significance threshold of $p < 0.05$ (unadjusted). SARTools[62] was used for EdgeR analysis and data visualization. Differentially enriched genomic regions identified from T47D and CD4[+] T cell ATAC-seq peak based analyses were merged to create a unified peak set. This peak set was later used in EdgeR analysis, applying an adjusted $P$-value threshold (FDR < 0.05) to distinguish between breast cancer patients and healthy donors. GREAT analysis[63] was used to assign the nearest genes located within 100 kbp to differentially enriched genomic loci. DAVID bioinformatics platform was used to obtain GO and KEGG pathways associated with differential enriched genomic regions[64]. Enrichment analysis was performed using DAVID's default human genome background, and significance was assessed using Benjamini–Hochberg-adjusted $p$-values (Benjamini values). To visualize the most significantly enriched terms, heatmaps were generated by a Python code. DAVID output files containing two columns, pathway name and Benjamini value, were imported and processed. Benjamini values were transformed using the -$\log_{10}$ scale to enhance interpretability. For both GO and KEGG analyses, the top 25-30 enriched terms (ranked by lowest Benjamini values) were selected for plotting. Color intensity reflects the transformed Benjamini value for each term. Box-and-whisker plots were generated using Python's matplotlib library to visualize read counts associated with genes in identified pathways[65]. In comparison between breast cancer ($n = 5$) and healthy donor ($n = 6$), a two-sided unpaired Student's t-test was used to assess the statistical significance. GraphPad Prism was used for statistical analysis, including cfDNA fragment size distribution analysis and open chromatin enrichment analysis.

## Peak annotation and genomic feature classification

Annotation of cfDNA enriched loci at ATAC-seq peaks was performed using the ChIPpeakAnno package in R[66]. Peaks were first converted to GRanges objects and annotated to the nearest genes using the TxDb.Hsapiens.UCSC.hg19.knownGene reference. Overlaps were determined using annotatePeakInBatch with output = "overlapping" and maxgap = 0. To characterize the genomic distribution of peaks, we used the genomicElementDistribution function. Regions were categorized into promoter, gene body, downstream, and distal intergenic elements. Promoter regions were defined as $-2000$ bp upstream to $+500$ bp downstream of the transcription start site (TSS), and subdivided into: Upstream 1–2 kb, Upstream 0.5-1 kb, Upstream < 500 bp, TSS ± 500 bp. Downstream regions were defined as 0–2000 bp beyond the transcription end site (TES), and gene body regions included all gene coding regions excluding promoters and downstream

ends. Distal intergenic regions were defined as those lacking overlap with any annotated gene features. In this analysis, intergenic peaks were treated as a single category and were not subdivided further. Additionally, peaks overlapping gene coding regions were further classified by genomic element type, including exons, introns, and intergenic regions. Exonic peaks were subcategorized into 5′ UTR, 3′ UTR, coding sequences (CDS), and other exons, based on transcript annotations from TxDb.Hsapiens.UCSC.hg19.knownGene, without using fixed length cutoffs. Classification was determined by direct overlap between peaks and annotated gene elements retrieved using fiveUTRsByTranscript, threeUTRsByTranscript, and cdsBy functions within the ChIPpeakAnno framework. Results were visualized as a three-level nested pie chart, representing promoter level, gene level, and exon level distributions.

## Multi-dimensional scaling (MDS) plot analysis

To visualize global variation in cfDNA fragment counts across selected genomic regions (1 kbp windows) between cancer and healthy donor samples, Multi-Dimensional Scaling (MDS) plot analysis was performed using the plotMDS function from the edgeR package within the SARTools framework or ggplot. The analysis was based on normalized read counts with the trimmed mean of M-values (TMM) method. MDS was computed using the top differentially enriched loci (adjusted $p < 0.05$), and distances between samples were calculated based on the leading log-fold change (leading logFC), defined as the root-mean-square average of the largest absolute $\log_2$ fold changes across the most variable features. The resulting axes represent the primary dimensions of variation in these logFC values. Sample clusters were visualized as ellipses using geom_mark_ellipse to represent grouping by condition. Ellipse boundaries were expanded slightly (3–5%) using the expand parameter for improved clarity.

## Volcano plot

To visualize differentially enriched cfDNA loci between cancer (breast or pancreatic) and healthy donor samples, volcano plots were generated using Python's matplotlib library. Input data were derived from EdgeR output and included genomic coordinates, $\log_2$ fold changes, and unadjusted $p$-values for each region. Significantly differentially enriched loci were defined as those with $< 0.05$ (unadjusted). Regions with $\log_2$ fold change $>0$ were classified as increased cfDNA enrichment loci, while those with $\log_2$ fold change $<0$ were considered decreased cfDNA enrichment loci.

## Heatmap of selected cfDNA enriched loci

Heatmaps were generated to visualize cfDNA enrichment patterns of differentially enriched loci associated with breast or pancreatic cancer related genes. Differentially enriched genomic regions were first identified using edgeR and then annotated to the nearest gene within 100 kb. To avoid redundancy, only one genomic region per gene (the region with the lowest adjusted p-value) was retained. Genes were sorted by ascending $\log_2$ fold change prior to plotting. Heatmaps were generated using the seaborn and matplotlib libraries in Python. Color intensity reflects normalized cfDNA fragment counts.

## Clinical data analysis

Kaplan–Meier survival analysis was performed using the Kaplan–Meier Plotter tool[67,68]. Patients were stratified into high and low EMP3 expression groups using the tool's auto cutoff algorithm, and survival was evaluated over a 60-month follow-up period. The log-rank p-value, hazard ratio (HR) with 95% confidence interval, and group sizes were automatically calculated. All patient samples were included.

Co-expression analysis between EMP3 and S100A4 mRNA levels was performed using mRNA expression data derived from the Illumina HT-12 v3 microarray platform accessed through cBioPortal for Cancer Genomics[69–71]. All samples from the dataset were included under default parameters with no filtering applied.

## Peripheral blood mononuclear cell (PBMC) isolation

PBMCs in the blood samples obtained from luminal breast cancer patients were purified by Ficoll gradient separation (Ficoll-Paque Plus, Cytiva). 7 mL of Ficoll-Paque gradient was pipetted into a 15-mL centrifuge tubes. The blood (7 mL) was diluted at a ratio of 1:1 in saline solution and carefully layered over the Ficoll-Paque gradient. The tubes were centrifuged for 30 min at $600 \times g$. The cell interface layer was harvested carefully, and the cells were washed in saline solution for 5 min at 3000 rpm. PBMCs were resuspended in 5 mL phosphate-buffered saline (PBS). Cells were counted using a TC20 Automated Cell Counter (Bio-Rad) and trypan blue.

## Fluorescence-activated cell sorting (FACS)

PBMCs were isolated by FACS. All PBMCs were washed, counted and suspended in ice-cold PBS, and cell-surface antigen staining was subsequently performed. Monoclonal antibodies against CD3 (APC), CD4 (PE/CYanine7, Pacific Blue), CD8 (PerCP/CYanine5.5), CD25 (APC), CD45RA (FITC) and Foxp3 (PE, eBioscience™ Foxp3/Transcription Factor Staining Buffer Set) were used in immunofluorescence staining. The stained cells were analyzed by BD FACSVerse (BD, USA), and data were analyzed using FlowJo v10.10. Gating was performed sequentially as follows: first, lymphocytes were identified based on forward scatter (FSC) and side scatter (SSC) properties. Singlets were then gated using FSC-A versus FSC-H, followed by exclusion of dead cells based on FSC/SSC parameters. From the live singlet population, $CD3^+$ T cells were identified, which were subsequently divided into $CD4^+$ and $CD8^+$ subsets. Within the $CD4^+$ compartment, $CD25^+Foxp3^+$ regulatory T cells were further quantified. Representative gating strategies are shown in Supplementary Fig. 3.

## XGBoost model development and prediction analysis

For the analysis of public cfDNA data, whole-genome cfDNA sequencing datasets from healthy donors and breast cancer patients were obtained from FinaleDB (Fragmentation Analysis of Cell-free DNA Database), incorporating data from three studies: Jiang et al., 2015 (EGA accession number EGAD00001001275); Adalsteinsson et al., 2017 (dbGaP accession code phs001417.v1.p1); and Cristiano et al., 2019 (EGA accession number EGAD00001005339)[15,23,72]. Additional cfDNA datasets from pancreatic and ovarian cancer patients were retrieved from the Cristiano et al. cohort via the European Genome-phenome Archive (EGA; accession number EGAD00001005339). A total of 182 cfDNA samples were analyzed, comprising 57 healthy donor samples, 64 breast cancer samples, 34 pancreatic cancer samples, and 27 ovarian cancer samples. For the pancreatic cancer and ovarian cancer data analyses, 50 healthy donor samples were extracted from Cristiano et al., 2019 (EGA accession number EGAD00001005339) data cohort. To classify cancer and healthy cfDNA profiles, we implemented supervised learning using the XGBoost classifier (xgboost v2.0.3) for all breast, pancreatic, and ovarian cancer models. Input features were generated using multiBigwigSummary from the deepTools suite (v3.5.6), which was used to extract normalized signal values from .bigWig files across selected genomic regions 3.5.1[73]. was employed to calculate average genomic scores with predefined resolution of, converting BigWig signals into numerical values compatible with machine learning analysis (10 kb bin). These regions included whole-genome, differentially enriched peaks from EdgeR analyses, or ATAC-seq peaks from $CD4^+$ T cells, T47D, PANC1, or PEO1. BED files specifying these loci were used as input to generate feature matrices, and the output files were used directly in machine learning models. For consistency across datasets, ATAC-seq peaks from PEO1 cells were converted from hg38 to hg19 using the UCSC LiftOver tool. An XGBoost[48] model was trained using cfDNA signals, either from the whole-genome or selected open chromatin regions. To generate a background set of control regions for comparative analysis, a random set of genomic intervals matched in number and size to the significant peaks was used. These 2804 random 10 kb regions were generated using shuffleBed (from BEDTools[74]) with the -noOverlapping flag to avoid overlapping regions. The 2804

differential peaks were used as a template bed file, and the randomization was constrained to mappable autosomal regions by excluding mitochondrial (chrM), unplaced (chrUn), random (chr_random), and sex chromosome Y (chrY) entries from the genome definition file (hg19_chrominfo.txt). For classification, datasets were split into training (70%) and test sets (30%). The XGBoost models were trained using objective = 'binary:logistic', max_depth=6, and eval_metric = 'aucpr' (area under the precision-recall curve), with early stopping applied (patience = 15 rounds) based on test set performance. Both randomized search optimization and hyperparameter grid search were considered (learning_rate: 0.05-0.65; reg_lambda: 1-10; scale_pos_weight: 0.8-1; n_estimators: 100; gamma: 0; tree_method: hist), with grid search specifically used to optimize the learning rate of the model. The validation score refers to the best precision-recall AUC achieved on the test set (clf_xgb.best_score) before early stopping. The accuracy was calculated using balanced accuracy, which averages recall across both classes and accounts for class imbalance, implemented via balanced_accuracy_score from scikit-learn (v1.3.2). Model performance was further assessed using ROC curves and AUC (area under the ROC curve) calculated from predicted class probabilities (roc_auc_score). Confusion matrices were constructed from predicted labels using ConfusionMatrixDisplay, and additional metrics including sensitivity, specificity, and precision were derived from true/false positives and negatives. To assess model stability, k-fold cross-validation (CV = 3–10) was conducted using xgboost.cv, with the highest mean AUC-PR score and corresponding boosting round recorded per fold. The best-performing hyperparameter set (based on top balanced accuracy and validation score) was selected for final evaluation and ROC/PR visualization. All visualizations (ROC, confusion matrix, feature importance) were created using matplotlib (v3.10.0), and all data processing was performed using pandas (v2.2.2) and numpy (v1.26.4).

## Ethics declarations

All ethical regulations relevant to human research participants were followed.

## Consent to participate

Human specimens used in this study were obtained from the Sanford Health Biobank and Chiba University, both of which follow established protocols for informed consent. All participants provided written informed consent for the use of their biological materials and associated data in research. The study was conducted in accordance with all applicable regulations governing confidentiality and data protection.

## Consent to publish

The samples and associated data were collected under institutional ethical approval from the Sanford Health Biobank and Chiba University. Donors provided informed consent permitting the use of their samples in scientific research and publication. All data were de-identified to ensure participant confidentiality.

## Statistics and reproducibility

All statistical analyses, including differential enrichment testing and machine learning evaluations, were performed using established bioinformatics tools and are described in the relevant sections of the Methods. Sample sizes, statistical tests, significance thresholds, and measures of variability are specified within each analysis subsection or figure legend. All key findings were confirmed in independent biological samples or public datasets where applicable, and reproducibility was ensured through the use of multiple patient or donor samples as indicated.

## Data availability

Raw data and processed cfDNA data have been deposited in NCBI's Gene Expression Omnibus and are accessible under accession number GSE279542. The ATAC-seq data were previously obtained[60], and the raw

and processed data are available under the GSE99479 accession number (https://www.ncbi.nlm.nih.gov/geo/query/acc.cgi?acc=GSE99479).

## Code availability

All custom codes are available in the following GitHub link: https://github.com/TakakuLab/cfDNA.

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

## Acknowledgements

We gratefully acknowledge the University of North Dakota School of Medicine and Health Sciences for providing student stipend support. We also thank the UND Genomics Core Facility for their outstanding technical assistance and Sanford coordinators (Ms. Brianna Szyszka-Study, Mr. Andrew Brummond-Study, and other members) for their support in clinical sample collection. This study uses data generated by The Cancer Genomics Laboratory at The Johns Hopkins University School of Medicine Sidney Kimmel Comprehensive Cancer Center as reported by Cristiano, S. et al.[15]. This work was supported by the ND-ACES (EPSCoR) pilot grant award and the DaCCoTA pilot grant award. M.T. is a Research Scholar of the American Cancer Society (ACS Research Scholar Award, RSG-23-645952-01-DMC). The graphical abstract was created in BioRender. Takaku, M. (2025) https://BioRender.com/hgdl3p4.

## Author contributions

S.D.G. and M. Takaku designed the overall research and wrote the manuscript. S.D.G., A.N., R.N., and M.S. purified all cfDNA samples. S.D.G. and M. Takaku analyzed the sequencing data. N.A.B. assisted with the initial XGBoost analysis. B.G. supported the establishment of the data analysis platform. S.G., T.L., and C.G. contributed to clinical sample collection. M.Y. and M. Takada performed T-cell sorting. X.W. contributed to project design. All authors participated in writing and editing the manuscript and approved the final version.

## Competing interests

The authors declare no competing interests.
