## [Transparent Peer Review file · Communications Biology]

Open Chromatin Guided Interpretable Machine Learning Reveals Cancer-Specific Chromatin Features in Cell-free DNA

Corresponding Author: Dr Motoki Takaku

Version 0:

Reviewer comments:

Reviewer #1

(Remarks to the Author)

This paper is well-written. The authors used ATAC-seq data to guide the selection of cancer-specific genomic regions in cfDNA to improve cancer identification. I have two suggestions below:

1. The number of cancer types is limited, I hope the authors could expand their analyses to other cancer types in the dataset reported by Cristiano and colleagues.
2. What are the other fragmentomic features such motif diversity score (proposed in “plasma DNA end-motif profiling as a fragmentomic marker in cancer, pregnancy and transplation”) and fragmentation profiles (propose in “fragmentation landscape of cell-free DNA revealed by deconvolutional analysis of end motifs”) in the ATAC-seq enrichment peaks? Do these two fragmentomic features derived from ATAC-seq enrichment peaks also achieve improved performance?

Reviewer #2

(Remarks to the Author)

The authors report experimental and computational analysis of cfDNA from patients with breast and pancreatic cancer. They base their analysis on the calculation of the sequencing coverage in 10,000 bp windows, genome-wide or focused on ATAC-seq peaks reported previously. This manuscript is within the important area of the analysis of cfDNA for cancer diagnostics and fits the scope of this journal. From my reading of the current version of the manuscript, it needs the following improvements:

Major points:

- 1) The literature review in the introduction needs to be significantly expanded (currently only 35 literature references). The authors need to show to the reader, where exactly is the current method positioned in the context of other existing methods. This includes, but not limited to the explanation of the following history of this field:
 - 1.1) As far as I know, the first work that performed cfDNA analysis based on genome-wide occupancy in large genomic windows was Cristiano et al, 2019: <https://www.nature.com/articles/s41586-019-1272-6>.
 - 1.2) After this, the first work that reported cancer/noncancer classification based on the analysis of nucleosome occupancy in specific windows with differential occupancy, and visualised these with PCA as in current manuscript, was Piroeva et al, 2023: <https://genome.cshlp.org/content/33/10/1649.full#F7>.
 - 1.3) The idea of combining cfDNA analysis with ATAC-seq defined peaks was previously implemented in several papers, e.g. including these:
Taklifi et al., 2022: <https://www.nature.com/articles/s41598-022-14675-z>
Bae et al., 2023: <https://www.nature.com/articles/s41467-023-37768-3>
 - 1.4) There have been several recent application of the analysis of cfDNA-based nucleosome occupancy in genomic

windows. E.g., for 10,000 bp windows as in the current work it was done in Takahashi et al, 2024:
<https://www.biorxiv.org/content/10.1101/2024.06.02.597054v1.full>

2) The authors mention that they work with “cfDNA from cell lines”, among other data. It is important to clarify, what do the authors mean by “cfDNA from cell lines”. cfDNA is usually defined for body fluids such as blood plasma or urine. Cell lines do not have this. If some analogy to cfDNA was used, but in fact some other source of DNA was used for cell lines, this needs to be explained. Also, the methods section does not include any description of wet lab experiments with cell lines – this needs to be added.

3) When talking about PCA plots, the authors do not explain, what exactly has been used for PCA. Is it the occupancy of cfDNA averaged for a given 10000-bp window? This needs to be explained to the reader (both in the Results and Methods sections).

4) Since this is dealing with patient samples, the methods section should probably include the reference to the relevant Ethical Approval(s).

5) The methods section needs to include more detailed description of cfDNA extraction. Currently it mentions “SubXTM-Exo-DNA kit”, but does not explain who is the manufacturer of this kit. Also, it’s possible that the letters “TM” should be in the superscript. The procedure of plasma separation and handling and storage also needs to be described.

6) In the Results section explaining ML model development, the authors write, “For public cfDNA data analysis, whole-genome cfDNA sequencing data were obtained from FinaleDB (Fragmentation Analysis of Cell-free DNA Database). A total of 121 cfDNA samples were analyzed, comprising 57 healthy donor samples and 64 breast cancer samples.”

6a) It is not clear from this description, from which original study were these samples (a reference to a third-party database does not clarify this; exact references to the primary databases hosting the raw data need to be provided such as GEO/SRA/DBGaP/EGA need to be provided, including the dataset IDs in these databases, and the literature citations as well.

6b) It is not clear how these datasets were processed.

7) Figure 1a: Why are mononucleosomes shown at the gel band of ~300 bp (rather than typical ~150-160 bp)? In general, all bands seem to be shifted on this figure. Does it mean that the gel shows us the DNA after adding sequencing adapters? If so, this is confusing for the reader and needs to be explained in detail, providing the lengths of the adapters, etc.

8) Figure 1b suggests that the authors have performed low-path sequencing coverage. Indeed, the Methods section mentions 30 million reads per sample. This is a very small number for a typical analysis of nucleosome occupancy, and with this low sequencing coverage it is expected that most reads will be coming from open chromatin regions. Thus, the very large enrichment of cfDNA in ATAC-seq peaks observed by the authors may be the artefact of the low sequencing coverage. At a higher sequencing coverage, cfDNA may be still enriched in open chromatin regions, but not so strongly.

9) Figure 1c mentions “in vitro cfDNA”, but cfDNA is defined only for body fluids of the organism (also see point 2 above). Please clarify this terminology and add a detailed description to the figure legends and Methods.

10) Figure 1d shows a very shallow depletion of cfDNA profile at TSS (almost no depletion at all). This may be the artefact of the low sequencing coverage (see also point 8 above).

11) The visualisation of the model performance in Figure 5a and 5b make it difficult to come up to a conclusion about the relative performance of the models with different ATAC-seq based peaks. It would be good to add error bars to the graph, and may be think about an alternative graphical (or table) representation allowing more precise comparison between different models.

12) What is the rationale of using ATAC-seq peaks from CD4+ T cells? It is a very specific cell type that is neither the most abundant in blood, nor related to breast tissues in a direct way.

13) It would be good to add a paragraph to the Methods section detailing the statistics of ATAC-seq peaks used in the current study – how many peaks for each cell type and a typical peak width.

Minor points:

14) Figure 3: add a description of the colour code for each heatmap.

15) Figure 3 legend (and through out the manuscript): clarify the meaning of “increased regions” and “increased peaks”. Do you mean “regions with increased cfDNA occupancy”?

16) Figure 4b: “Diferencial” should be “Differential”.

17) Figure 5: Explain the meaning of the “validation score”

Reviewer #3

(Remarks to the Author)

This manuscript presents an innovative approach to analyzing cfDNA fragments in both cancer and healthy donor samples guided by ATAC-seq analysis. The authors demonstrate that nucleosomal DNA fragments from breast and pancreatic cancer patients show distinct enrichment patterns at open chromatin regions, and they successfully leverage this pattern using XGBoost to enhance cancer detection accuracy.

While the study presents promising findings and advances our understanding of cfDNA-based cancer detection, there are several areas that would benefit from additional clarification and analysis. I have outlined these points below.

Comment 1

The similarity in fragment size distributions between breast cancer and healthy samples shown in Figure 1a and Supplementary Figure 1b is unexpected, given that tumor-derived cfDNA typically exhibits much shorter fragment lengths. This observation might be attributed to low tumor fractions in the breast cancer samples. We recommend including additional clinical metadata, cfDNA concentration measurements, and tumor fraction estimates (ichorDNA) to better contextualize these findings.

Additionally, we suggest incorporating Supplementary Figure 1b into the main text, as it provides a more quantitative visualization of fragment size distributions compared to the TapeStation images.

Comment 2

There appears to be a discrepancy regarding the differential peak analysis methodology. While edgeR is referenced on page 6, the methods section (page 13) describes a DESeq2-based workflow without mentioning edgeR. Please clarify which tool was ultimately used for the analysis.

Comment 3

If edgeR was indeed employed for differential peak identification, the choice of normalization method requires careful consideration. Both edgeR's TMM and DESeq2's RLE normalization assume that most features remain unchanged between conditions. This assumption may not hold for cfDNA analyses. We recommend investigating whether the hypothesis holds, or evaluating alternative normalization strategies and perhaps developing a cfDNA-specific approach that incorporates cfDNA concentration and tumor fraction.

Comment 4

The methods section should specify the background set used for GO and KEGG pathway enrichment analyses. Given that the analysis was restricted to peaks defined from CD4+ T and T47D ATAC-seq data, the choice of background is crucial for avoiding false discoveries.

Comment 5

To complement the current GO and KEGG pathway enrichment analyses, we recommend including GSEA, which may provide more robust insights for this study.

Comment 6

The clustering patterns observed in the PCA plots (Figures 3e and 4f) should be evaluated for potential batch effects, as such separation is commonly observed in tumor vs healthy gene expression data, especially when they come from different batches. We suggest implementing batch correction methods and establishing baselines, in order to demonstrate that the observed classification power indeed stems from the selected genomic loci.

Comment 7

The term "leading logFC" in the PCA plots requires clarification. Please specify which signals or values were used for the principal component transformation.

Version 1:

Reviewer comments:

Reviewer #1

(Remarks to the Author)

The authors have addressed all my concerns.

Reviewer #2

(Remarks to the Author)

The authors have addressed my suggestions

Reviewer #3

(Remarks to the Author)

I have carefully reviewed the manuscript and the author's response to my previous comments. I'm glad to see that the authors have addressed all my concerns. I think this manuscript makes a solid contribution to the field of liquid biopsy and cancer early detection.

We would like to thank all the reviewers for their constructive and insightful comments. We have carefully addressed each concern and revised the manuscript accordingly. We believe these revisions have substantially improved the quality and clarity of the work. Below, we provide our point-by-point responses to each comment. All changes made to the main text are highlighted in blue.

Reviewers' comments:

Reviewer #1 (Remarks to the Author):

This paper is well-written. The authors used ATAC-seq data to guide the selection of cancer-specific genomic regions in cfDNA to improve cancer identification. I have two suggestions below:

We appreciate the reviewer's positive remarks regarding our manuscript and the thoughtful suggestions provided. Below, we address each comment in detail.

1. The number of cancer types is limited, I hope the authors could expand their analyses to other cancer types in the dataset reported by Cristiano and colleagues.

We appreciate this constructive suggestion. To further expand our study, we obtained pancreatic (N=34) and ovarian (N=27) cancer cfDNA data from Cristiano et al data cohort. Using the same XGBoost machine learning (ML) pipeline developed for breast cancer analysis, we performed ML-based prediction analyses for each cancer type.

For pancreatic cancer, the highest prediction accuracy was achieved when pancreatic cancer-specific enhancer and promoter regions were used (PC Model 2). In contrast to breast cancer, incorporating CD4⁺ T cell open chromatin regions resulted in only marginal improvement or, in some validation methods, no improvement (PC Model 3). Interestingly, for ovarian cancer cfDNA (at least within the Cristiano et al. dataset), the highest prediction accuracy was observed when using whole-genome cfDNA enrichment data (Ovarian Cancer Whole Genome Model). The inclusion of either ovarian cancer-specific (OC Model 2) or CD4⁺ T cell-specific (OC Model 2) open chromatin regions did not improve prediction performance and, in fact, yielded lower accuracy scores. These findings suggest that cfDNA fragmentation patterns may exhibit cancer type-specific characteristics, and that leveraging open chromatin regions does not universally enhance the detection of cancer-derived cfDNA fragments. The results from this expanded analysis have been added to Supplementary Figure 6e and Supplementary Table 3. We have also revised the Discussion section to incorporate these new findings.

2. What are the other fragmentomic features such motif diversity score (proposed in “plasma DNA end-motif profiling as a fragmentomic marker in cancer, pregnancy and transplation”) and fragmentation profiles (propose in “fragmentation landscape of cell-free DNA revealed by deconvolutional analysis of end motifs”) in the ATAC-seq enrichment peaks? Do these two fragmentomic features derived from ATAC-seq enrichment peaks also achieve improved performance?

We appreciate this insightful suggestion. To address this point, we analyzed the DNA fragment end motifs in our cfDNA dataset and examined whether motif enrichment differs at breast cancer-specific open chromatin regions. Consistent with previous reports, we observed enrichment of the CCNN motif. However, the degree of enrichment was similar between breast cancer and healthy donor samples at open chromatin regions. This may reflect the relatively low proportion of tumor-derived DNA fragments, as noted in another reviewer comment. Our copy number variation analysis indicated a limited fraction of tumor-derived cfDNA. Given these findings, we did not pursue further ML-based prediction using motif features. Nonetheless, the observed motif enrichment supports the quality of our cfDNA data and its consistency with prior studies. These results have been added to Supplementary Figure 1e-f.

Reviewer #2 (Remarks to the Author):

The authors report experimental and computational analysis of cfDNA from patients with breast and pancreatic cancer. They base their analysis on the calculation of the sequencing coverage in 10,000 bp windows, genome-wide or focused on ATAC-seq peaks reported previously. This manuscript is within the important area of the analysis of cfDNA for cancer diagnostics and fits the scope of this journal. From my reading of the current version of the manuscript, it needs the following improvements:

We appreciate the reviewer's summary of our study and the recognition of its significance within the field of cfDNA-based cancer diagnostics.

Major points:

1) The literature review in the introduction needs to be significantly expanded (currently only 35 literature references). The authors need to show to the reader, where exactly is the current method positioned in the context of other existing methods. This includes, but not limited to the explanation of the following history of this field:

1.1) As far as I know, the first work that performed cfDNA analysis based on genome-wide occupancy in large genomic windows was Cristiano et al, 2019: <https://www.nature.com/articles/s41586-019-1272-6>.

1.2) After this, the first work that reported cancer/noncancer classification based on the analysis of nucleosome occupancy in specific windows with differential occupancy, and visualised these with PCA as in current manuscript, was Piroeva et al, 2023: <https://genome.cshlp.org/content/33/10/1649.full#F7>.

1.3) The idea of combining cfDNA analysis with ATAC-seq defined peaks was previously implemented in several papers, e.g. including these:

Taklifi et al., 2022: <https://www.nature.com/articles/s41598-022-14675-z>

Bae et al., 2023: <https://www.nature.com/articles/s41467-023-37768-3>

1.4) There have been several recent application of the analysis of cfDNA-based nucleosome occupancy in genomic windows. E.g., for 10,000 bp windows as in the current work it was done in Takahashi et al, 2024: <https://www.biorxiv.org/content/10.1101/2024.06.02.597054v1.full>

We appreciate this suggestion and fully agree that our original citation list was somewhat limited. In response, we have incorporated the recommended references, along with additional relevant studies, primarily in the Introduction and Discussion sections. We would also like to highlight that differential enrichment was observed not only at cancer specific open chromatin regions but also at T cell specific open chromatin regions. Importantly, the inclusion of T cell specific open chromatin data further improved the performance of our machine learning prediction for breast cancer. These novel findings are emphasized in the revised manuscript.

2) The authors mention that they work with “cfDNA from cell lines”, among other data. It is important to clarify, what do the authors mean by “cfDNA from cell lines”. cfDNA is usually defined for body fluids such as blood plasma or urine. Cell lines do not have this. If some analogy to cfDNA was used, but in fact some other source of DNA was used for cell lines, this needs to be explained. Also, the methods section does not include any description of wet lab experiments with cell lines – this needs to be added.

We apologize for the lack of description in the original manuscript. To recapitulate *in vivo* (blood-derived) cfDNA, the supernatant from cultured cell media has been used in several previous studies (References 36-39). These DNA fragments are generally considered to be released through apoptosis and other cell death processes. This *in vitro* approach is valuable for elucidating the fundamental biology of cfDNA release directly from tumor cells, as it eliminates confounding factors such as immune cell responses and other complex biological processes involved in cfDNA generation *in vivo*. In the revised manuscript, we have clarified this rationale in the Results section and have added a detailed description of the experimental procedures to the Methods section.

The revised Results section now reads:

To characterize the fundamental patterns of cfDNAs released from cancer cells, we first collected *in vitro* cfDNA from luminal breast cancer cell lines (T47D and KPL-1). Cells were cultured to confluence, after which the culture medium was collected and carefully filtered to remove cellular debris (see Methods). This *in vitro* approach enables the isolation of cfDNA fragments released directly from cancer cells, without the confounding influence of immune cell contributions or enzymatic DNA degradation that typically occurs in the bloodstream.

3) When talking about PCA plots, the authors do not explain, what exactly has been used for PCA. Is it the occupancy of cfDNA averaged for a given 10000-bp window? This needs to be explained to the reader (both in the Results and Methods sections).

We apologize again for the lack of clarity in our original descriptions. PCA (or MDS) plots were generated using normalized read counts at selected (differential) open chromatin regions. We have revised the descriptions in the Results section, figure legends, and Methods section to clearly explain this analytical approach. To clarify, the signal intensities used for the plot were based on read counts at ATAC-seq peaks, each extended to a 1 kb window. Log₂ fold change values obtained from the **EdgeR** analysis were used as input. We have updated the relevant descriptions in the Results, Methods, and figure legends to reflect this approach.

The updated Methods section:

Multi-dimensional scaling (MDS) plot analysis

To visualize global variation in cfDNA fragment counts across selected genomic regions (1 kb windows) between cancer and healthy donor samples, Multi-Dimensional Scaling (MDS) plot analysis was performed using the plotMDS function from the edgeR package within the SARTools framework or ggplot. The analysis was based on normalized read counts with the

trimmed mean of M-values (TMM) method. MDS was computed using the top differentially enriched loci (adjusted $p < 0.05$), and distances between samples were calculated based on the leading log-fold change (leading logFC), defined as the root-mean-square average of the largest absolute \log_2 fold changes across the most variable features. The resulting axes represent the primary dimensions of variation in these logFC values. Sample clusters were visualized as ellipses using `geom_mark_ellipse` to represent grouping by condition. Ellipse boundaries were expanded slightly (3–5%) using the `expand` parameter for improved clarity.

4) Since this is dealing with patient samples, the methods section should probably include the reference to the relevant Ethical Approval(s).

We fully agree with this important point. The revised manuscript now includes comprehensive ethical approval statements, including Ethics Declarations, Consent to Participate, and Consent to Publish, in accordance with journal guidelines.

5) The methods section needs to include more detailed description of cfDNA extraction. Currently it mentions “SubXTM-Exo-DNA kit”, but does not explain who is the manufacturer of this kit. Also, it’s possible that the letters “TM” should be in the superscript. The procedure of plasma separation and handling and storage also needs to be described.

We greatly appreciate this comment. In response, we have updated the Methods section to include additional details regarding our cfDNA purification procedures.

6) In the Results section explaining ML model development, the authors write, “For public cfDNA data analysis, whole-genome cfDNA sequencing data were obtained from FinaleDB (Fragmentation Analysis of Cell-free DNA Database). A total of 121 cfDNA samples were analyzed, comprising 57 healthy donor samples and 64 breast cancer samples.”

6a) It is not clear from this description, from which original study were these samples (a reference to a third-party database does not clarify this; exact references to the primary databases hosting the raw data need to be provided such as GEO/SRA/DBGaP/EGA need to be provided, including the dataset IDs in these databases, and the literature citations as well.

We apologize for the lack of descriptions. We included the details in the revised Methods section. In short, we used three data sets (Jiang et al., 2015; Adalsteinsson et al., 2017; Cristiano et al., 2019).

6b) It is not clear how these datasets were processed.

We apologize for the insufficient description. In the previous version of the FinaleDB website, cfDNA genome coverage data were directly available for download in BigWig file format. However, following a recent update, these files are no longer accessible, and the authors of the original database were unable to share them directly. To address this issue and in response to Reviewer 1’s suggestion, we independently processed Cristiano et al. cfDNA data from

pancreatic cancer, ovarian cancer, and healthy donors to generate genome coverage profiles compatible with our machine learning analysis. We have added a detailed description of this updated processing workflow to the Methods section.

7) Figure 1a: Why are mononucleosomes shown at the gel band of ~300 bp (rather than typical ~150-160 bp)? In general, all bands seem to be shifted on this figure. Does it mean that the gel shows us the DNA after adding sequencing adapters? If so, this is confusing for the reader and needs to be explained in detail, providing the lengths of the adapters, etc.

We completely agree with this comment. The TapeStation image shows sequencing libraries, which include adapter sequences (~100 bp), resulting in a shift in fragment size. To avoid confusion, we have revised the descriptions in the Results section and the corresponding figure legends to clearly indicate this point.

8) Figure 1b suggests that the authors have performed low-path sequencing coverage. Indeed, the Methods section mentions 30 million reads per sample. This is a very small number for a typical analysis of nucleosome occupancy, and with this low sequencing coverage it is expected that most reads will be coming from open chromatin regions. Thus, the very large enrichment of cfDNA in ATAC-seq peaks observed by the authors may be the artefact of the low sequencing coverage. At a higher sequencing coverage, cfDNA may be still enriched in open chromatin regions, but not so strongly.

We appreciate this thoughtful suggestion. We would first like to clarify that the primary motivation of our study is to establish a cost-effective analytical pipeline for cfDNA-based cancer detection. While the fundamental mechanisms underlying cfDNA enrichment at open chromatin regions remain incompletely understood, this pattern may result from a combination of biological processes and technical or methodological biases. For example, it is well known that NGS library preparation protocols favor the amplification of short DNA fragments, and that DNA fragments from closed or heterochromatic regions may be underrepresented due to inefficient recovery or amplification during cfDNA purification and library construction.

To address this concern, we collected deep sequencing data from three breast cancer samples.
Sample 1: ~117 million uniquely mapped reads without duplicates (~313 million raw reads)
Sample 2: ~132 million uniquely mapped reads without duplicates (~254 million raw reads)
Sample 3: ~36 million uniquely mapped reads without duplicates (~139 million raw reads)

Metaplot analysis confirmed that cfDNA fragments remained enriched at T47D ATAC-seq peaks to a degree comparable to that observed in the shallow sequencing dataset. These results support the robustness of our conclusion, independent of sequencing depth. The new results have been added to Supplementary Figure 1d.

9) Figure 1c mentions “in vitro cfDNA”, but cfDNA is defined only for body fluids of the organism (also see point 2 above). Please clarify this terminology and add a detailed description to the figure legends and Methods.

As discussed earlier, we have revised the definition and updated the descriptions accordingly.

10) Figure 1d shows a very shallow depletion of cfDNA profile at TSS (almost no depletion at all). This may be the artefact of the low sequencing coverage (see also point 8 above).

We appreciate the reviewer’s observation. To address this concern, we repeated the metaplot analysis using deep sequencing data (>100 million reads). One of the examples is shown below. The results confirmed that the nucleosome enrichment pattern remained consistent between shallow and deep sequencing datasets, indicating that the observed profiles are not an artifact of sequencing depth. While the enrichment of the -1 nucleosome appears somewhat weaker, the $+1$ nucleosome downstream of the TSS is clearly enriched. As discussed earlier, these differences may be attributed to our cfDNA purification protocol. Additionally, we observed typical symmetric nucleosome enrichment patterns at enhancer regions, further supporting the integrity of our cfDNA data.

11) The visualisation of the model performance in Figure 5a and 5b make sit difficult to come up to a conclusion about the relative performance of the models with different ATAC-seq based peaks. It would be good to add error bars to the graph, and may be think about an alternative graphical (or table) representation allowing more precise comparison between different models.

While it is challenging to add error bars in this graph as the sample numbers are limited for ML analysis, we have added Supplementary Table 3 summarizing the ML results.

12) What is the rationale of using ATAC-seq peaks from CD4⁺ T cells? It is a very specific cell type that is neither the most abundant in blood, nor related to breast tissues in a direct way.

The rationale for using CD4⁺ T cell ATAC-seq peaks is supported by both existing literature and our own data. Although the proportion of CD4⁺ T cells can vary between individuals and health conditions, our FACS analysis demonstrated clear enrichment of CD4⁺ T cells among peripheral blood immune cells in breast cancer patients. Since a substantial fraction of cfDNA is thought to originate from hematopoietic cells, including lymphocytes, we reasoned that incorporating open chromatin regions specific to CD4⁺ T cells could help capture differentially enriched cfDNA regions that distinguish breast cancer patients from healthy individuals. Indeed, our EdgeR differential enrichment analysis and machine learning predictions both confirmed that significant cfDNA signal differences are present at CD4⁺ T cell specific open chromatin regions. These findings support the inclusion of this cell type in our analytical framework.

Additionally, our FACS data also revealed the presence of CD8⁺ T cells in peripheral blood. Given the known role of CD8⁺ T cells in anti-tumor immune responses, we believe that including CD8⁺ T cell specific chromatin profiles in future analyses may provide further insights into immune related cfDNA signatures in cancer patients. However, evaluating CD8⁺ T cell contributions would require longitudinal sampling (e.g., before and after surgery or treatment), which is beyond the scope of the current study. We also anticipate a substantial overlap between CD8⁺ T cells and CD4⁺ T cell open chromatin regions.

The newly generated FACS data have been added to Supplementary Figure 3, and we have revised the Results section to clarify the rationale behind the use of CD4⁺ T cell ATAC-seq peaks.

13) It would be good to add a paragraph to the Methods section detailing the statistics of ATAC-seq peaks used in the current study – how many peaks for each cell type and a typical peak width.

We appreciate this suggestion. We have revised the Methods section to include additional details regarding the ATAC-seq peaks used in this study, including the number of peaks and the peak width. For differential peak analysis, all peaks were extended to 1 kbp.

Minor points:

14) Figure 3: add a description of the colour code for each heatmap.

We have added the color code.

15) Figure 3 legend (and through out the manuscript): clarify the meaning of “increased regions”

and “increased peaks”. Do you mean “regions with increased cfDNA occupancy”?

We have revised the figure legends and added the detailed descriptions. Specifically, we now refer to “increased regions” or “increased peaks” as “regions with increased cfDNA occupancy, enrichment, or levels”.

Figure 3 legend now reads:

Figure 3. Integrative analysis of cfDNA enrichment using CD4⁺ T cell and breast cancer ATAC-seq peaks.

a. Differential enrichment analysis at CD4⁺ T cell ATAC-seq peaks. The volcano plot shows differential signals between breast cancer and healthy donor cfDNAs. Regions significantly (p -values <0.05) upregulated and downregulated are marked in red and blue, respectively. **b.** Peak annotation analysis of differentially enriched regions. Differentially enriched loci are categorized into promoter, exon, intron, or intergenic regions. **c.** KEGG pathway enrichment analysis of genes associated with differential loci based on CD4⁺ T cell ATAC-seq peak analysis. **d.** Overlap of differential peaks. Venn diagram depicts the intersection of differential peaks identified by T47D or CD4⁺ T cell ATAC-seq peak based analysis. A total of 2,804 peaks are contained in the merged list. **e.** PCA plot demonstrating distinct clustering of cfNuc signals between healthy and breast cancer samples, based on 2,804 selected genomic regions. **f.** Distribution of 2,804 union peaks relative to TSS. **g.** Heatmap displaying normalized cfDNA read counts for genes associated with the top differentially enriched regions. Read counts were derived from differential peaks and mapped to their nearest genes. Color intensity reflects relative read count values. **h.** KEGG pathway enrichment analysis of genes associated with regions of increased cfDNA occupancy. Pathway analysis was performed using genes linked to genomic regions showing higher cfDNA enrichment in breast cancer samples. **i.** Heatmap displaying normalized cfDNA read counts for genes associated with the KEGG breast cancer pathway. Each column represents an individual sample, and each row corresponds to a gene. Color intensity reflects the relative level of cfDNA enrichment at genomic regions linked to each gene.

16) Figure 4b: “Diferencial” should be “Differential”.

We have corrected those errors.

17) Figure 5: Explain the meaning of the “validation score”

We apologize for the lack of detail. We have revised the Methods section to clarify how the validation score was calculated, and updated the description in the Figure 5 legend accordingly.

Updated Methods section:

XGBoost model development and prediction analysis

For the analysis of public cfDNA data, whole-genome cfDNA sequencing datasets from healthy donors and breast cancer patients were obtained from FinaleDB (Fragmentation Analysis of Cell-free DNA Database), incorporating data from three studies: Jiang et al., 2015 (EGA accession number EGAD00001001275); Adalsteinsson et al., 2017 (dbGaP accession code

phs001417.v1.p1); and Cristiano et al., 2019 (EGA accession number EGAD00001005339)^{15, 23, 72}. Additional cfDNA datasets from pancreatic and ovarian cancer patients were retrieved from the Cristiano et al. cohort via the European Genome-phenome Archive (EGA; accession number EGAD00001005339). A total of 182 cfDNA samples were analyzed, comprising 57 healthy donor samples, 64 breast cancer samples, 34 pancreatic cancer samples, and 27 ovarian cancer samples. For the pancreatic cancer and ovarian cancer data analyses, 50 healthy donor samples were extracted from Cristiano et al., 2019 (EGA accession number EGAD00001005339) data cohort. To classify cancer and healthy cfDNA profiles, we implemented supervised learning using the XGBoost classifier (xgboost v2.0.3) for all breast, pancreatic, and ovarian cancer models. Input features were generated using multiBigwigSummary from the deepTools suite (v3.5.6), which was used to extract normalized signal values from .bigWig files across selected genomic regions 3.5.1⁷³. was employed to calculate average genomic scores with predefined resolution of, converting BigWig signals into numerical values compatible with machine learning analysis (10 kb bin). These regions included whole genome, differentially enriched peaks from EdgeR analyses, or ATAC-seq peaks from CD4⁺ T cells, T47D, PANC1, or PEO1. BED files specifying these loci were used as input to generate feature matrices, and the output files were used directly in machine learning models. An XGBoost⁴⁹ model was trained using cfDNA signals, either from the whole genome or selected open chromatin regions. To generate a background set of control regions for comparative analysis, a random set of genomic intervals matched in number and size to the significant peaks was used. These 2,804 random 10 kb regions were generated using shuffleBed (from BEDTools⁷⁴) with the -noOverlapping flag to avoid overlapping regions. The 2,804 differential peaks was used as a template bed file, and the randomization was constrained to mappable autosomal regions by excluding mitochondrial (chrM), unplaced (chrUn), random (chr_random), and sex chromosome Y (chrY) entries from the genome definition file (hg19_chrominfo.txt). For classification, datasets were split into training (70%) and test sets (30%). The XGBoost models were trained using objective='binary:logistic', max_depth=6, and eval_metric='aucpr' (area under the precision-recall curve), with early stopping applied (patience = 15 rounds) based on test set performance. Both randomized search optimization and hyperparameter grid search were considered (learning_rate: 0.05-0.65; reg_lambda: 1-10; scale_pos_weight: 0.8-1; n_estimators: 100; gamma: 0; tree_method: hist), with grid search specifically used to optimize the learning rate of the model. The validation score refers to the best precision-recall AUC achieved on the test set (clf_xgb.best_score) before early stopping. The accuracy was calculated using balanced accuracy, which averages recall across both classes and accounts for class imbalance, implemented via balanced_accuracy_score from scikit-learn (v1.3.2). Model performance was further assessed using ROC curves and AUC (area under the ROC curve) calculated from predicted class probabilities (roc_auc_score). Confusion matrices were constructed from predicted labels using ConfusionMatrixDisplay, and additional metrics including sensitivity, specificity, and precision were derived from true/false positives and negatives. To assess model stability, k-fold cross-validation (CV = 3–10) was conducted using xgboost.cv, with the highest mean AUC-PR score and corresponding boosting round recorded per fold. The best-performing hyperparameter set (based on top balanced accuracy and validation score) was selected for final evaluation and ROC/PR visualization. All visualizations (ROC, confusion matrix, feature importance) were created using matplotlib

(v3.10.0), and all data processing was performed using pandas (v2.2.2) and numpy (v1.26.4).

Reviewer #3 (Remarks to the Author):

This manuscript presents an innovative approach to analyzing cfDNA fragments in both cancer and healthy donor samples guided by ATAC-seq analysis. The authors demonstrate that nucleosomal DNA fragments from breast and pancreatic cancer patients show distinct enrichment patterns at open chromatin regions, and they successfully leverage this pattern using XGBoost to enhance cancer detection accuracy.

While the study presents promising findings and advances our understanding of cfDNA-based cancer detection, there are several areas that would benefit from additional clarification and analysis. I have outlined these points below.

We thank the reviewer for their thoughtful and constructive summary of our work.

Comment 1

The similarity in fragment size distributions between breast cancer and healthy samples shown in Figure 1a and Supplementary Figure 1b is unexpected, given that tumor-derived cfDNA typically exhibits much shorter fragment lengths. This observation might be attributed to low tumor fractions in the breast cancer samples. We recommend including additional clinical metadata, cfDNA concentration measurements, and tumor fraction estimates (ichorDNA) to better contextualize these findings.

Additionally, we suggest incorporating Supplementary Figure 1b into the main text, as it provides a more quantitative visualization of fragment size distributions compared to the TapeStation images.

We appreciate this important question and valuable suggestion. While shorter tumor-derived cfDNA fragments have been reported in previous studies, such differences are often more pronounced when using single-stranded DNA library preparation methods. In our study, we used a magnetic bead-based protocol that first isolate the exosome fraction and subsequently purify cfDNAs. A magnetic bead-based protocol is known to preferentially retain longer DNA fragments. Consistent with the previous literature, our data showed a substantial presence of di-nucleosome fragments, in addition to mono-nucleosomes. Furthermore, in a separate study (the manuscript is currently under review), we analyzed cfDNA from advanced breast cancer patients and observed even longer fragments, including tri-nucleosome-sized DNA. Our library preparation protocol is based on a standard double-stranded DNA approach, which likely favors the retention of more intact cfDNA fragments.

To provide additional context, we have included a clinical metadata table for each patient sample. We also estimated tumor fractions using the ichorCNA tool, which yielded tumor content estimates ranging from approximately 1% to 3%. As detailed in the table, all of our

cfDNA samples were collected at early disease stages, and many of them were collected after surgery and/or treatment, which may explain the relatively low abundance of tumor-derived DNA based on copy number–based inference.

Given these observations, we agree with the reviewer that the proportion of tumor-derived cfDNA may be limited in our cohort. This likely contributes to the overall similarity in size distribution between cfDNA from breast cancer patients and healthy donors. The low tumor fraction may also explain why the inclusion of T cell specific open chromatin regions was effective in capturing differentially enriched regions and distinguishing breast cancer cfDNA signatures from those of healthy individuals. We have added the clinical metadata table (Supplementary Table 1) and the ichorCNA analysis (Supplementary Figure 1e-f) results to the revised manuscript. We have also updated relevant sections of the Results and Discussion, and we have moved Supplementary Figure 1b to the main figure panel (Figure 1c).

Comment 2

There appears to be a discrepancy regarding the differential peak analysis methodology. While edgeR is referenced on page 6, the methods section (page 13) describes a DESeq2-based workflow without mentioning edgeR. Please clarify which tool was ultimately used for the analysis.

We apologize for the discrepancy in our descriptions. All differential enrichment analyses were conducted using EdgeR. We have revised the Results, Figures, Figure Legends, and Methods sections to ensure consistency and eliminate any confusion regarding the analysis tools used.

Comment 3

If edgeR was indeed employed for differential peak identification, the choice of normalization method requires careful consideration. Both edgeR's TMM and DESeq2's RLE normalization assume that most features remain unchanged between conditions. This assumption may not hold for cfDNA analyses. We recommend investigating whether the hypothesis hold, or evaluating alternative normalization strategies and perhaps developing a cfDNA-specific approach that incorporates cfDNA concentration and tumor fraction.

We appreciate this important comment. Initially, we observed differential enrichment of cfDNA at specific loci through visual inspection using genome browser tracks and by performing box plot analyses at selected regions. These box plots were based on simple read depth normalization, which differs from the statistical normalization methods used in EdgeR and DESeq2. Additionally, metaplot analyses at selected open chromatin regions indicated that average signal intensities were broadly similar between cfDNA samples from breast cancer patients and healthy donors. These observations suggested that overall cfDNA enrichment patterns are generally consistent across conditions. To further validate our findings, we conducted a complementary differential analysis using DESeq2. The results showed that the majority of

differential peaks identified by DESeq2 overlapped with those detected by EdgeR, reinforcing the robustness of our conclusions.

Comment 4

The methods section should specify the background set used for GO and KEGG pathway enrichment analyses. Given that the analysis was restricted to peaks defined from CD4+ T and T47D ATAC-seq data, the choice of background is crucial for avoiding false discoveries.

We appreciate this comment and have added the relevant background information to the revised manuscript. For our pathway enrichment analysis, we used the entire human genome (~20,000 protein-coding genes) as the background set. As noted in our response to Comment 5, even if we were to restrict the background to genes associated with ATAC-seq peaks, the number of reference genes would still be approximately 10,000, roughly half of all protein-coding genes. Thus, we believe our current approach remains a reasonable and widely accepted practice for enrichment analysis. We acknowledge that using the full genome as a background could introduce bias toward pathways associated with the ATAC-seq source cell types (e.g., breast cancer or CD4⁺ T cells). However, our results include enrichment of diverse biological pathways that extend beyond those specific to the input cell types, suggesting that the analysis captures biologically meaningful and relevant pathways.

Comment 5

To complement the current GO and KEGG pathway enrichment analyses, we recommend including GSEA, which may provide more robust insights for this study.

We appreciate the reviewer's suggestion to include GSEA as a complementary analysis to GO and KEGG enrichment. One of the key challenges in GSEA analysis is the assignment of peaks to genes. In response, we performed GSEA using genes associated with 51,463 T47D ATAC-seq entire peaks. Among these peaks, 29,642 genes were located within 100 kb of the peaks, based on our standard peak-to-gene assignment criteria that we used for our downstream analysis. To generate a non-redundant input for GSEA, we selected the peak with the highest absolute fold-change for each gene and removed duplicates, resulting in a list of 10,000 unique genes. However, this analysis did not yield any pathways that met the FDR < 0.05 significance threshold. We believe this may be due to limitations in the treatment of duplicated genes and the use of fold-change values that do not account for sample variability. Given these results, we believe our original pathway analysis approach (based on assigning differential peaks to their nearest genes followed by GO and KEGG enrichment) provides an effective framework for identifying enriched pathways in our cfDNA data.

Comment 6

The clustering patterns observed in the PCA plots (Figures 3e and 4f) should be evaluated for potential batch effects, as such separation is commonly observed in tumor vs healthy gene

expression data, especially when they come from different batches. We suggest implementing batch correction methods and establishing baselines, in order to demonstrate that the observed classification power indeed stems from the selected genomic loci.

We would first like to emphasize that the MDS (labeled as PCA in the earlier version) plots presented in our study were generated using cfDNA data specifically at differentially enriched genomic regions. To minimize technical variability, all cfDNA purification was performed concurrently, and all sequencing libraries were processed on the same lane. While the timing of plasma collection varied slightly within and across sample groups, all samples were prepared following the same standardized procedures. Therefore, we do not believe that the observed separation patterns in the MDS plots are attributable to batch effects. As previously discussed, differential enrichment was observed only at specific genomic loci. Furthermore, we applied various normalization strategies, including read depth normalization, EdgeR, and DESeq2, all of which consistently confirmed site-specific differential enrichment patterns. Based on these findings, we believe the observed differences reflect true biological variation in cfDNA enrichment rather than technical artifacts.

Comment 7

The term "leading logFC" in the PCA plots requires clarification. Please specify which signals or values were used for the principal component transformation.

We apologize for the confusion and insufficient description. We agree that the term "leading logFC" requires clarification, and it may have also contributed to the concerns raised in the previous comment. In our analysis, "leading logFC" refers to the \log_2 fold-change values of the most differentially enriched genomic regions identified by EdgeR. These values were used to perform dimensionality reduction and generate the MDS plots (labeled as PCA in earlier versions). The transformation was based on normalized read counts at selected, differentially enriched genomic loci between groups. We have revised the figure legends and Methods section to clearly define this term and explain how the principal components (or dimensions) were derived.